# Personalized functional brain network topography is associated with individual differences in youth cognition

Arielle S. Keller [1,2], Adam R. Pines[1,2], Sheila Shanmugan[1,2], Valerie J. Sydnor [1,2], Zaixu Cui [3], Maxwell A. Bertolero [1,2], Ran Barzilay[2,4,5], Aaron F. Alexander-Bloch [2,4,5], Nora Byington[6], Andrew Chen [7], Gregory M. Conan[6], Christos Davatzikos [8], Eric Feczko[6], Timothy J. Hendrickson [6,9], Audrey Houghton [6], Bart Larsen[1,2], Hongming Li[8], Oscar Miranda-Dominguez [6], David R. Roalf [2,5], Anders Perrone[6], Alisha Shetty[1,2], Russell T. Shinohara[7], Yong Fan[8], Damien A. Fair [6,10] & Theodore D. Satterthwaite [1,2,4,10] ✉

Individual differences in cognition during childhood are associated with important social, physical, and mental health outcomes in adolescence and adulthood. Given that cortical surface arealization during development reflects the brain's functional prioritization, quantifying variation in the topography of functional brain networks across the developing cortex may provide insight regarding individual differences in cognition. We test this idea by defining personalized functional networks (PFNs) that account for inter-individual heterogeneity in functional brain network topography in 9–10 year olds from the Adolescent Brain Cognitive Development℠ Study. Across matched discovery ($n = 3525$) and replication ($n = 3447$) samples, the total cortical representation of fronto-parietal PFNs positively correlates with general cognition. Cross-validated ridge regressions trained on PFN topography predict cognition in unseen data across domains, with prediction accuracy increasing along the cortex's sensorimotor-association organizational axis. These results establish that functional network topography heterogeneity is associated with individual differences in cognition before the critical transition into adolescence.

Individual differences in cognition during childhood are associated with academic performance[1] and quality of life in youth[2], as well as social, physical and mental health outcomes in adulthood[3–5]. Moreover, cognitive deficits during youth are associated with heightened risk for psychopathology[6], risk-taking behaviors[7], cardiovascular disease[8–10], and all-cause mortality[11,12]. Understanding how individual differences in cognitive functioning emerge during childhood is a critical prerequisite for efforts that seek to promote healthy neuro-cognitive development. Prior neuroimaging studies have demonstrated that complex cognitive tasks engage spatially-distributed, large-scale association networks[13–15]. However, less is known about the relationship between individual differences in cognition and the spatial layout of functional networks on the anatomic cortex—an individual's functional topography. Attempts at investigating this important problem have faced two key challenges. First, methods must account for person-specific variation in functional topography across individuals, which is especially pronounced in association cortices[16]. Second, recent studies have emphasized that reproducible

brain-behavior associations may require very large samples[17]. We seek to overcome these challenges by capitalizing upon recent advances in machine learning to identify individual-specific functional brain networks in large discovery and replication samples. We test the overarching hypothesis that the functional topography of association networks is associated with individual differences in cognitive function in children.

Studies in humans using fMRI have typically studied functional brain networks using a "one-size-fits-all" approach with standardized network atlases[18,19]. In this approach, a 1:1 correspondence between structural and functional neuroanatomy across individuals is assumed, as fMRI data is co-registered to a structural image and then normalized to a structural template. This critical assumption has been proven to be demonstrably false by studies from multiple independent laboratories[20–23]. These studies have revealed substantial interindividual heterogeneity in functional topography[20–25], with especially notable heterogeneity in networks in association cortex that support higher-order cognition and are implicated in cognitive impairments in psychiatric illness in adults[21,26]. To overcome this challenge, precision functional mapping techniques have been developed as an alternative to using group-level atlases. These techniques are used to derive individually-defined networks that capture each brain's unique pattern of functional topography. Such personalized functional networks (PFNs) have been found to be highly stable within individuals and to predict an individual's spatial pattern of activation on fMRI tasks[21,22,27].

Notably, the same networks that both support higher-order cognition and have the greatest variability in functional topography tend to lie near the upper end of a predominant axis of hierarchical cortical organization known as the sensorimotor-association (S-A) axis, which spans from unimodal visual and somatomotor cortex to transmodal association cortex[28]. The S-A axis summarizes the canonical spatial patterning of numerous cortical properties, including myelination, evolutionary expansion, transcriptomics, metabolism, and the principal gradient of functional connectivity[29]. Prominent individual variation in the functional topography of networks at the association pole—including the fronto-parietal network, ventral attention network, and default mode network—has been posited to impact individual differences in cognition[23]. Indeed, our collaborative group[16] recently reported that greater total cortical representation of fronto-parietal PFNs was associated with better cognitive performance, and found that a model trained on the complex pattern of functional topography could predict cognition in unseen data. However, while these results were drawn from a large study, it was collected at a single site, and has not yet been replicated. This limitation points to the ongoing challenge of reproducibility in studies that seek to define brain-behavior relationships in humans. The reproducibility crisis has been documented extensively[30,31], marked by failed replications of high-profile findings[32,33], and has prompted a renewed emphasis on methods to increase the generalizability of computational models to new datasets[34]. In addition to the well-documented problems arising from small sample sizes[17] and over-fitting[35], it may also be the case that a lack of consideration for individual-specific neuroanatomy has also contributed to weak effect sizes and non-reproducible findings of prior work.

Here, we aim to delineate the relationship between functional topography and individual differences in cognition by conducting a replication and extension of ref. 16 in two large, matched samples of youth from the Adolescent Brain Cognitive Development℠ (ABCD) Study[36–38] (total $n = 6972$). Using spatially-regularized non-negative matrix factorization (NMF)[39], we identify personalized functional brain networks (PFNs) that capture inter-individual heterogeneity in functional topography while maintaining interpretability. We seek to replicate two key results[16]. First, we aim to demonstrate that predictive models trained on PFN topography can predict youth cognition in unseen data. Second, we aim to replicate the finding that fronto-parietal PFN topography is associated with individual differences in cognition. Furthermore, we aim to extend prior work by investigating whether PFN topography is predictive of general or specific cognitive abilities by training models to predict three major domains of cognition[40] (general cognition, executive function, and learning/memory). Finally, we predict that the strength of associations between functional topography and cognition will align with the cortical hierarchy defined by the S-A axis, with the functional topography of PFNs in association cortex yielding the most accurate predictions of individual differences in cognition. As described below, this study constitutes the largest replication of precision functional mapping in children to date, identifying reproducible brain-behavior associations and demonstrating that these relationships align with a major cortical hierarchy.

## Results

We aimed to understand how individual differences in functional brain network topography relate to individual differences in cognitive functioning in a sample of $n = 6972$ children aged 9–10 years old from the ABCD Study. To account for inter-individual heterogeneity in the spatial layout of functional brain networks, we used precision functional mapping to define PFNs for each individual. Leveraging a previously-defined group atlas[16], we used an advanced machine learning method – spatially-regularized NMF—to identify 17 PFNs within each individual (Fig. 1). This procedure yielded a set of 17 matrices of network weights across each vertex (soft parcellation; used for primary analyses) as well as a matrix of non-overlapping networks describing the highest network weight at each vertex (hard parcellation; used for secondary analyses and visualization). To determine where each PFN fell along a predominant axis of cortical hierarchical organization, we computed the average S-A axis rank across the vertices within each PFN using the group-averaged hard parcellation.

### PFN topography predicts individual differences in cognition in unseen data

We first sought to replicate the prior finding that the multivariate pattern of PFN topography could predict cognitive performance in unseen data. Here, general cognition was operationalized as the first principal component from a Bayesian probabilistic principal components analysis (BPPCA) computed in a prior study, capturing the largest amount of variance across nine cognitive tasks[40]. As previously[16], we trained ridge regression models using the cortical representation of each PFN (network loadings at each vertex) while controlling for age, sex, site, and head motion. Leveraging our matched discovery and replication samples for out-of-sample testing, we first trained models in the discovery sample using nested cross-validation for parameter tuning, and then used the held-out replication sample for testing. We then performed the opposite procedure, performing nested training in the replication sample and testing in the held-out discovery sample. We found that individualized functional topography accurately predicted out-of-sample cognitive performance in both samples (Fig. 2a, discovery: $r(3525) = 0.41$, $p < 0.001$, 95% CI: [0.39, 0.44]; replication: $r(3447) = 0.45$, $p < 0.001$, 95% CI: [0.43, 0.48]), with effect sizes at the higher end of the expected range from predictive modeling studies using functional connectivity in prior work[41–43]. Demonstrating that our results were not dependent on the matched discovery and replication sample split, we also applied repeated random cross-validation over one hundred repetitions as previously[16], which returned similar results (Fig. 2b, mean $r = 0.44$, $p < 0.001$, 95% CI: [0.43, 0.44]). These results show high consistency with correlations between actual and predicted cognitive performance reported in prior work[16] (Matched sample 1: $r = 0.46$, $p < 0.001$; Matched sample 2: $r = 0.41$, $p < 0.001$; Repeated random CV: mean $r = 0.42$, $p < 0.001$; see Fig. 7 in ref. 16). Given that many prior studies in this dataset have demonstrated associations between socio-economic status and cognitive functioning[44–53], we also note that our predictive models trained on PFN topography

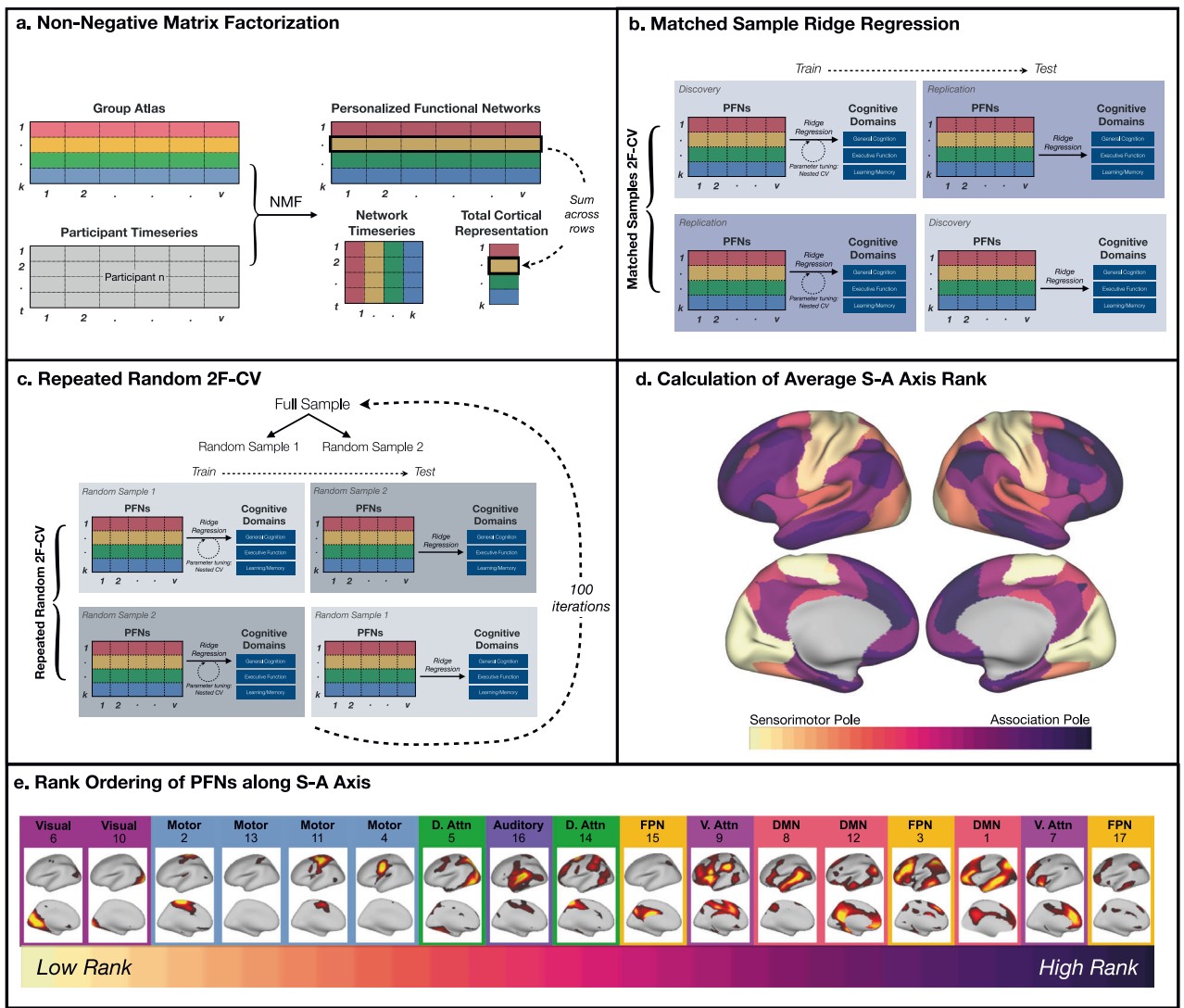

**Fig. 1 | Identification and analysis of Personalized Functional Networks (PFNs).**
**a** Using a previously-defined group atlas[16] as a prior, we generated personalized functional networks (PFNs) by applying non-negative matrix factorization (NMF) to each individual participant's vertex by time matrix. This procedure allows each network in the consensus group atlas to have a varying cortical representation in each individual, thereby capturing individual differences in the size and layout of networks while simultaneously allowing for interpretable between-individual comparisons. We also calculated the total cortical representation of each PFN by summing each network's loadings across all vertices. **b** To evaluate whether an individual's multivariate pattern of PFN topography could accurately predict cognition in unseen data, we trained linear ridge regression models using the cortical representation of each PFN while controlling for age, sex, site, and head motion. Leveraging our matched discovery and replication samples for two-fold cross-validation (2F-CV), we first trained models in the discovery sample using nested cross-validation for parameter tuning, and then tested these models in the held-out replication sample. We then performed nested training in the replication sample and testing in the held-out discovery sample. **c** To demonstrate that our results were not dependent on the matched discovery and replication sample split, we conducted repeated random cross-validation over one hundred iterations, each time performing a random split of our full sample and applying two-fold cross-validation. **d** Next, we calculated the average sensorimotor-association (S-A) axis rank across the vertices contained within each PFN. **e** We then rank-ordered each PFN according to its average S-A rank. Brain maps depict vertex loadings for each PFN (D. Attn Dorsal Attention, V. Attn Ventral Attention, DMN Default Mode Network, FPN Fronto-Parietal Network).

outperformed models trained on socio-economic status as measured by areal deprivation index (ADI) alone (Supplementary Table 1) and have separately characterized the multidimensional features of childhood environments that are reflected in PFN topography[54].

To evaluate the relative contributions of each network to prediction accuracy, we trained linear ridge regression models on the functional topography of each PFN independently. We found that the fronto-parietal and ventral attention networks tended to have the highest prediction accuracies, whereas the somatomotor and visual networks tended to have the lowest (Fig. 2c, d). These results are consistent with the feature weights from models which used all features and align with our prior report[16] (see Supplementary Fig. 1 for

exact replication), with consistency in prediction accuracies across datasets and samples. Together, these results suggest that individual variation in functional network topography has important implications for cognitive performance in youth.

## PFN topography predicts executive function and learning/memory with reduced accuracy

We next evaluated whether multivariate patterns of PFN topography could be used to predict cognitive performance in held-out data across other cognitive domains. We again trained linear ridge regression models using PFN topography and identical covariates to predict either executive function or learning/memory, which are the second and third

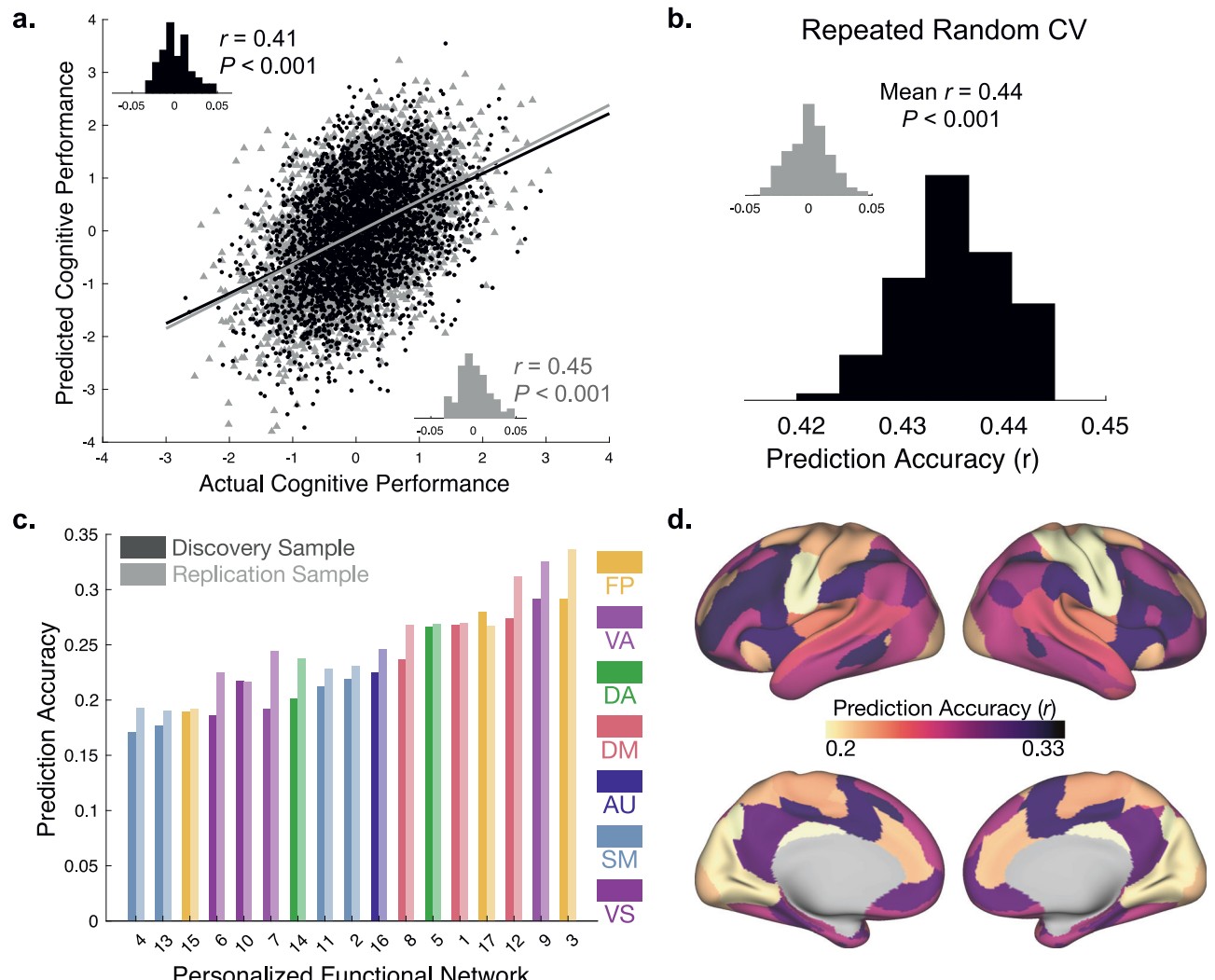

**Fig. 2 | Functional topography of association networks predicts individual differences in general cognition in unseen data. a** Association between actual and predicted cognitive performance using two-fold cross-validation (2F-CV) with nested cross-validation for parameter tuning across both the discovery (black scatterplot; $r(3525) = 0.41$, $p = 3.050 \times 10^{-146}$) and replication (gray scatterplot; $r(3447) = 0.45$, $p = 3.850 \times 10^{-174}$) samples. Inset histograms represent the distributions of prediction accuracies from a permutation test. **b** Repeated random 2F-CV (100 runs) provided evidence of stable prediction accuracy across splits of the data, which was far better than a null distribution with permuted data (inset). Two-sided $t$-test reveals that repeated random 2F-CV prediction accuracies are significantly greater than the null distribution of prediction accuracies with permuted data

$(t(100) = 261.274, p = 2.595 \times 10^{-253})$. **c** Prediction accuracy is shown for seventeen models trained on each PFN independently for the discovery sample (dark bars) and replication sample (light bars), with the highest prediction accuracies found in the ventral attention and fronto-parietal control networks. Note that all $p$-values associated with prediction accuracies are significant after Bonferroni correction for multiple comparisons. (FP Fronto-Parietal, VA Ventral Attention, DA Dorsal Attention, DM Default Mode, AU Auditory, SM Somatomotor, VS Visual). **d** Functional topography within association networks yield the most accurate predictions of general cognition. Prediction accuracy across the full sample shown for seventeen cross-validated models trained on each PFN independently.

ranked principal components capturing variance across nine cognitive tasks[40]. We hypothesized that general cognition would show stronger associations with functional topography than secondary or tertiary cognitive domains. Notably, while the first cognitive accuracy factor from our prior report[16] is typically referred to as "executive function and complex cognition" (and abbreviated as "executive function"), it most aligns with the general cognition factor from the ABCD Study[55].

It is worth noting that while these prediction accuracies were less strong than for the first principal component of general cognition, we found that individualized functional topography predicted performance in our two samples for both executive function (Fig. 3a, discovery: $r(3525) = 0.17$, $p < 0.001$, 95% CI: [0.14, 0.20]; replication: $r(3447) = 0.16$, $p < 0.001$, 95% CI: [0.13, 0.20]) and learning/memory (Fig. 3e, discovery: $r(3525) = 0.27$, $p < 0.001$, 95% CI: [0.24, 0.30]; replication: $r(3447) = 0.27$, $p < 0.001$, 95% CI: [0.24, 0.30]). Repeated

random two-fold cross-validation again returned similar results (Fig. 3b, mean $r = 0.17$, $p < 0.001$, 95% CI: [0.17, 0.17]; Fig. 3f, mean $r = 0.28$, $p < 0.001$, 95% CI: [0.28, 0.28]). When ridge regression models were trained using the topography of each PFN independently, fronto-parietal and ventral attention PFNs yielded the highest prediction accuracies for both executive function (Fig. 3c, d) and learning/memory (Fig. 3d, h). Associations between actual and predicted cognitive performance across all three tasks are depicted as hexplots in Supplementary Fig. 2 to visualize the density of points given the large number of participants in this study.

**Links between functional topography and cognition align with a network's position in the cortical hierarchy**

Motivated by our observation that fronto-parietal association network topography contributed most to the prediction of cognitive

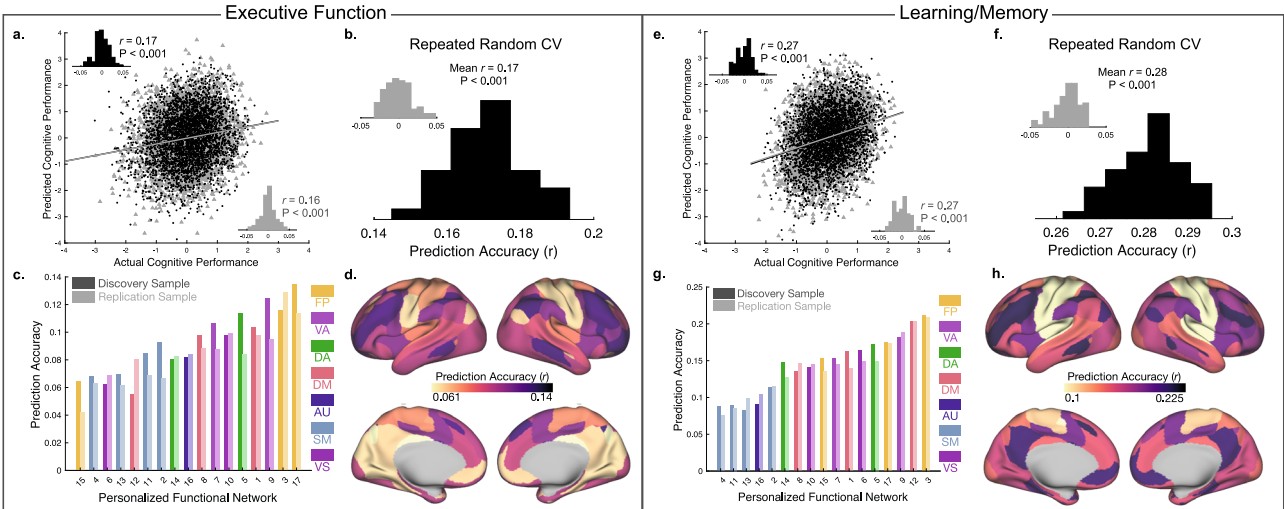

**Fig. 3 | Functional topography of association networks predicts individual differences in multiple cognitive domains in unseen data.** Results of ridge regression models predicting individual differences in executive function (**a**–**d**) and learning/memory (**e**–**h**). Panels a/e: Association between actual and predicted executive function (**a**) or learning/memory (**e**) using two-fold cross-validation (2F-CV) across both the discovery (black scatterplot) and replication (gray scatterplot) samples. Inset histograms represent the distributions of prediction accuracies from a permutation test. Repeated random 2F-CV (100 runs) provided evidence of stable prediction accuracy across many splits of the data for both executive function (**b**) and learning/memory (**f**), which was far better than a null distribution with

permuted data (inset). The PFNs with the highest prediction accuracies for executive function (**c**, **d**) and learning/memory (**g**, **h**) were found in association cortex and were maximal in the ventral attention and fronto-parietal control networks. Prediction accuracy is shown for seventeen models trained on each PFN independently for the discovery sample (dark bars) and replication sample (light bars) in (**c**, **g**). Note that all p-values associated with prediction accuracies are significant after Bonferroni correction for multiple comparisons. (FP Fronto-Parietal, VA Ventral Attention, DA Dorsal Attention, DM Default Mode, AU Auditory, SM Somatomotor, VS Visual).

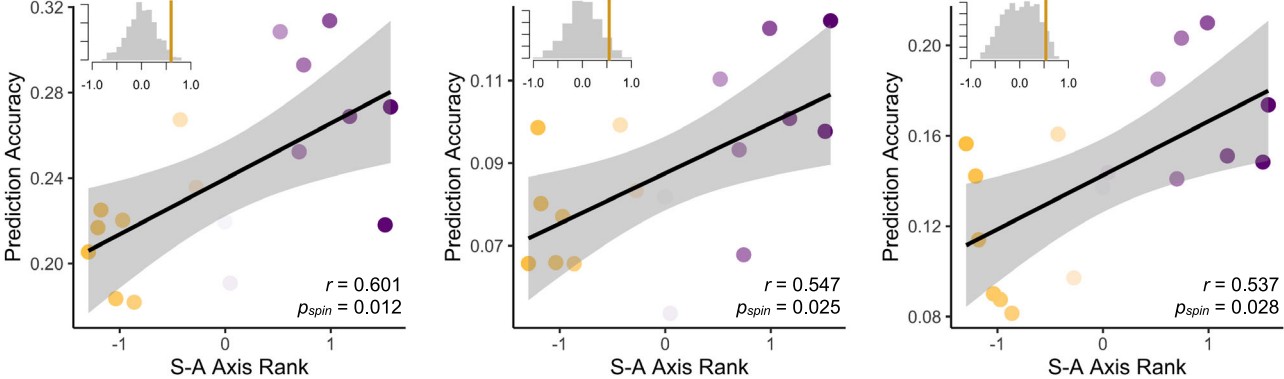

**Fig. 4 | Prediction accuracy of functional topography varies systematically along the S-A axis.** The sensorimotor-association (S-A) axis represents a hierarchy of cortical organization. The prediction accuracies of models trained on each PFN independently are significantly associated with the rank of each PFN along the S-A axis as shown by statistically significant Spearman correlations (two-sided) for the 17 networks across all three cognitive domains: general cognition (left), executive

function (middle), and learning/memory (right). Shaded gray error bands represent 95% confidence intervals. Note: average S-A axis ranks for each PFN are z-scored for visualization purposes. Inset histograms depict the distribution of Spearman correlations between rank and prediction accuracy for 1000 spin-based permutations of the S-A axis, with the vertical line showing the true Spearman correlation value.

performance while somatomotor networks contributed the least, we next investigated whether the predictive accuracy of a given network's ridge regression model was linearly associated with that network's rank along the S-A axis[28]. To account for the spatial auto-correlation of the data, we leveraged a widely-used spin-based spatial permutation procedure[56]. We found that prediction accuracy and position along the S-A axis were significantly correlated for predictions of general cognition (Spearman $r(17) = 0.601$, $p_{spin} = 0.012$) executive function (Spearman $r(17) = 0.547$, $p_{spin} = 0.025$) and learning/memory (Spearman $r(17) = 0.537$, $p_{spin} = 0.028$; Fig. 4). These results demonstrate that a network's position along the S-A axis is associated with the relevance of its functional topography in predicting cognitive performance in youth, providing a useful framework for describing and understanding the spatial pattern of prediction accuracy results across networks.

## The total cortical representation of fronto-parietal PFNs is associated with cognitive performance

We also sought to replicate previously-reported associations between the functional topography of individual PFNs and cognitive performance[16]. We previously found that greater cortical representations of two fronto-parietal networks (networks 15 and 17) were associated with better cognitive performance (see Figure 6 in ref. 16). As previously[16], we first calculated the total cortical representation of each PFN as the sum of network loadings across all vertices, using the soft parcellation to account for spatial overlap across functional brain networks. As described in prior work[16], this measure of total cortical representation captures the spatial extent of each PFN on the cortical surface. We then applied linear mixed-effects models to probe the association between total cortical representation of each PFN and

**Table 1 | Linear mixed effects models depicting associations between general cognition and fronto-parietal PFN topography**

| Predictors | PFN 3 | | | | PFN 15 | | | | PFN 17 | | | |
|---|---|---|---|---|---|---|---|---|---|---|---|---|
| | β | Std. Error | t | $p_{bonf}$ | β | Std. Error | t | $p_{bonf}$ | β | Std. Error | t | $p_{bonf}$ |
| **Discovery** | | | | | | | | | | | | |
| Intercept | 0.02 | 0.02 | 0.69 | 0.488 | 0.07 | 0.02 | 2.95 | **0.003** | −0.04 | 0.02 | −1.52 | 0.128 |
| Age | −0.04 | 0.02 | −2.54 | **0.011** | −0.02 | 0.02 | −1.28 | 0.201 | −0.04 | 0.02 | −2.58 | **0.010** |
| Sex | −0.05 | 0.03 | −1.60 | 0.109 | −0.15 | 0.03 | −4.51 | **$6.72 \times 10^{-6}$** | 0.05 | 0.03 | 1.57 | 0.117 |
| Mean FD | 0.12 | 0.02 | 6.98 | **$3.44 \times 10^{-12}$** | 0.12 | 0.02 | 6.82 | **$1.06 \times 10^{-11}$** | 0.04 | 0.02 | 2.22 | **0.027** |
| General Cognition | 0.08 | 0.02 | 3.24 | **0.001** | 0.09 | 0.02 | 3.50 | **$4.67 \times 10^{-4}$** | 0.10 | 0.02 | 4.29 | **$1.88 \times 10^{-5}$** |
| **Replication** | | | | | | | | | | | | |
| Intercept | 0.01 | 0.02 | 0.43 | 0.665 | 0.07 | 0.02 | 2.82 | **0.005** | −0.02 | 0.02 | −1.00 | 0.320 |
| Age | −0.01 | 0.02 | −0.84 | 0.400 | −0.06 | 0.02 | −3.32 | **$9.02 \times 10^{-4}$** | −0.05 | 0.02 | −2.65 | **0.008** |
| Sex | −0.04 | 0.03 | −1.28 | 0.199 | −0.15 | 0.03 | −4.27 | **$2.01 \times 10^{-5}$** | 0.04 | 0.03 | 1.11 | 0.265 |
| Mean FD | 0.16 | 0.02 | 9.05 | **$2.39 \times 10^{-19}$** | 0.08 | 0.02 | 4.54 | **$5.81 \times 10^{-6}$** | 0.04 | 0.02 | 2.11 | **0.035** |
| General Cognition | 0.08 | 0.02 | 3.07 | **0.002** | 0.09 | 0.02 | 3.74 | **$1.84 \times 10^{-4}$** | 0.12 | 0.02 | 4.68 | **$2.98 \times 10^{-6}$** |

Note that data were harmonized across sites using ComBat[58,59] and each model also included a random effect term for family ID. Bold font indicates statistically significant p-values ($p < 0.05$) after Bonferroni correction for multiple comparisons

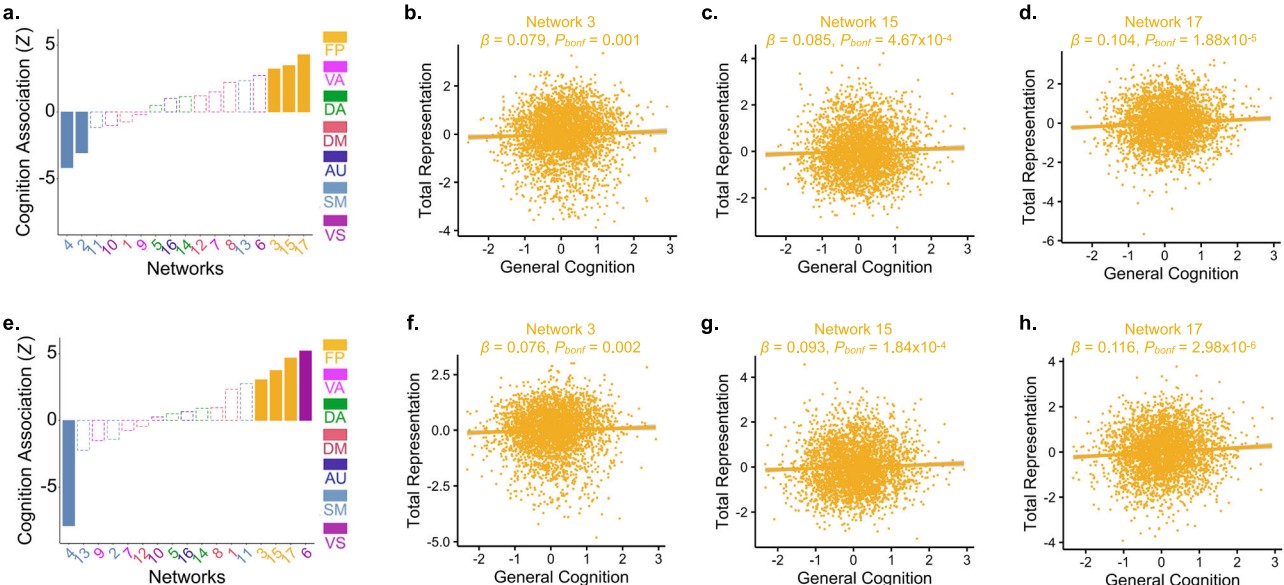

**Fig. 5 | Total cortical representations of fronto-parietal PFNs are positively associated with cognition.** Ordering the seventeen PFNs by the strength of their signed association with general cognition, we found significant positive associations between general cognition and the total cortical representation of all three fronto-parietal PFNs and negative correlations with a somatomotor network in both the discovery (**a–d**) and replication (**e–h**) samples ($P_{Bonf} < 0.05$; dashed lines indicate networks with effects that were not statistically significant). Scatterplots depict the relationship between general cognition and the total cortical representation of fronto-parietal networks 3, 15, and 17. (FP Fronto-Parietal, VA Ventral Attention, DA Dorsal Attention, DM Default Mode, AU Auditory, SM Somatomotor, VS Visual).

general cognition. These models accounted for age, sex (motivated by prior findings[57] that patterns of PFN total cortical representation differ by biological sex), family (to account for siblings in the ABCD dataset), and head motion (mean FD) as model covariates (Table 1; Fig. 5). Multiple comparisons were accounted for using the Bonferroni method. ComBat harmonization was applied to account for variability across sites[58,59].

We hypothesized that greater total cortical representation of association networks would be associated with better general cognitive abilities, in line with the intuition that more cortical surface space dedicated to these networks would facilitate the functions they subserve. We found that all three fronto-parietal PFNs (networks 3, 15, and 17) were significantly positively associated with general cognition

across both the discovery and replication samples. These findings remained significant in sensitivity analyses controlling for psychotropic medication use (Supplementary Table 2) as well as socioeconomic status (Supplementary Table 3). We also note that although the effect sizes for these univariate associations are small, they fall within or above the expected range for accurately-estimated brain-behavior effect sizes in studies of this size[17] and these effects are highly reproducible across studies and samples. Together, these results replicate the findings presented in Figure 6 of ref. 16. In addition to replicating these prior results regarding fronto-parietal network topography in two large samples, we additionally found that one somatomotor network (network 4) was inversely associated with cognitive performance in both discovery and replication samples, and

another somatomotor network (network 2) was inversely associated with cognition in only the discovery sample. Notably, the total cortical representation of network 2 was similarly found to be inversely related to cognition in the original report by ref. 16.

## Discussion

In the largest study to use precision functional brain mapping to investigate cognition in children to date, we found reproducible associations between individual differences in functional brain network organization and individual differences in cognition. Replicating key findings from a prior study[16] in samples that were an order of magnitude larger, we trained cross-validated models on the complex multivariate pattern of PFN topography to predict individual differences in cognitive functioning in unseen participants' data. Critically, we identified a consistent spatial pattern that accounts for these results, whereby association network topography yields the strongest predictions and sensorimotor network topography yields the weakest predictions of cognitive functioning, directly aligning with the S-A axis of hierarchical cortical organization[28]. Together, these findings demonstrate that the link between functional network topography and cognition in children on the precipice of the transition to adolescence is reproducible, representing an important step toward understanding heterogeneity in neurocognitive development.

### Scalable precision functional brain mapping in children

Our approach successfully overcame two key challenges: addressing inter-individual heterogeneity in functional brain network organization using precision functional mapping and addressing the need for reproducibility by developing cross-validated models in two large samples of thousands of individuals. The reproducibility crisis continues to pose a significant challenge for neuroscience and psychology research, with recent findings further emphasizing the need for very large sample sizes to uncover reproducible brain-wide associations with behavior[17]. However, large-scale open-source datasets such as the ABCD Study® provide significant hope for a feasible path forward. Our results represent the largest successful replication of associations between functional brain network topography and cognition in children, with consistent findings across datasets and samples. Several important factors are likely to have contributed to this success. First, both the original study[16] and our replication leveraged datasets with a large number of fMRI scans of children's brains ($n = 693$ in the original study, and $n = 6972$ in our replication). These datasets allowed us to uncover reliable associations between functional brain network topography and cognition. Second, our studies made use of predictive models that were trained and tested using rigorous cross-validation. Third, our precision functional brain mapping approach of identifying unique functional networks in individual children's brains allowed us to capitalize on inter-individual variability rather than treat such variability as noise. This approach may have contributed to the relatively larger effect sizes we observed compared with prior studies using group atlases, which may have bolstered our ability to reproduce these findings. In order to move toward the goal of supporting child and adolescent mental health, moderate to large effect sizes and generalizability to new samples are essential for predictive models to have clinical utility. Our scalable precision functional mapping approach may therefore be leveraged in future studies of children and adolescents to harness individual variability and identify important, reproducible brain-behavior associations.

### Association network topography supports domain-general cognitive abilities

Having replicated key brain network-cognition associations from prior work[16], we can begin to interpret these associations in the context of brain development. Our observation that general cognitive abilities are more strongly associated with PFN topography than other cognitive domains suggests that greater spatial representation of association networks across the cortex may support domain-general cognitive abilities. This finding builds upon prior results highlighting the predominance of general cognitive abilities (also referred to as a g-factor[60,61]) in accounting for shared variance across cognitive tasks. Recent work in the ABCD dataset has highlighted the potential role of this g-factor in mediating between genetic risk and psychopathology in children[46], suggesting that our identification of functional topography patterns associated with general cognition may represent a brain feature of interest for future studies of resilience.

We also found a similar pattern across all three cognitive domains in terms of which PFNs most strongly contributed to predictions of cognitive performance. This consistent pattern was well-described by a major hierarchy of cortical organization known as the S-A axis, with association networks contributing the most to associations with and predictions of cognitive performance across domains. This finding provides further evidence for the existence of a perception-cognition processing hierarchy in the brain[29,62] that aligns with the S-A axis[28]. Indeed, the association networks whose topography was most strongly associated with cognitive performance in children also show the greatest evolutionary expansion between non-human primates and humans[63,64] and their function has been correlated with cognitive performance in adults[13–15]. Prior studies have also demonstrated that this S-A axis gradually becomes the predominant pattern of cortical functional organizational with age, as the principal gradient of functional connectivity shifts from a visuo-motor axis to the S-A axis from childhood to adolescence[65]—a shift which happens during the protracted development of higher-order cognitive functions[62]. Future studies may investigate how longitudinal developmental changes in PFN functional topography along the S-A axis are related to the maturation of complex cognitive abilities, complementing cross-sectional work[16,66].

Another potential explanation for why variability in functional topography in the association cortices is the most strongly associated with individual differences in cognition is that these regions, and particularly regions of the fronto-parietal network, also tend to have the highest degree of inter-individual heterogeneity in other features[16,19,67–69]. Thus, while various networks across the S-A axis likely contribute in diverse ways to cognitive functioning, the notable individual variability in association network topography may be a more salient feature for predicting individual differences in cognition. Indeed, these regions tend to exhibit lower structural and functional heritability[70,71] and undergo the greatest surface area expansion during development[63]. Moreover, the extended window of plasticity for these regions compared with other parts of the cortex[72] renders them more likely to be shaped by an individual's environment and experiences[71], potentially further contributing to their unique spatial patterning across individuals. Encouragingly, this extended window in which association networks remain plastic may also indicate that interventions targeting these systems could be effective in supporting the development of healthy cognition.

### Limitations and future directions

This study had several limitations worth noting. First, this study was conducted at a single timepoint, using the baseline cohort from the ABCD Study®. As such, we were able to train models that could predict cognitive performance from functional brain network topography in held-out participants data, but were not able to build predictive models of future changes in cognition within an individual. Our work therefore sets the foundation for future longitudinal studies using the ABCD Study® dataset to identify changes in functional brain network organization during development using the PFNs we have identified at this baseline assessment. Second, head motion continues to pose an ongoing challenge for neuroimaging studies[73–75], particularly for studies of children[73], and indeed our linear mixed effects models revealed

that the total cortical representation of fronto-parietal PFNs was significantly associated with head motion. We have attempted to mitigate these effects in our analyses of interest by following best practices for reducing the influence of head motion on our results, including using a top-performing preprocessing pipeline and inclusion of motion as a covariate in all analyses. Third, we used data combined across four fMRI runs, including three where a behavioral task was performed during scanning, in line with prior studies of PFNs[16,66], aiming to maximize the amount of high-quality data for our study. Prior studies have shown that variation in functional networks is primarily driven by inter-individual heterogeneity rather than task-related factors and that intrinsic functional networks are similar during task and rest[68]. Fourth, differences between the ABCD dataset and the dataset used in the original study[16] (e.g., differences in scanning sequences, registration templates, and cognitive measures) prohibited us from directly applying the same models from the original study to this dataset directly. Our results therefore constitute a conceptual replication of the prior findings that demonstrates the robust generalizability of the results with both new data and new methods. Fifth, the registration of MRI data to a common reference space (*fslr*) has known limitations (e.g., the spatial warping to register individual brains with this common reference will necessarily differ across individuals). While this approach had the advantage of allowing us to apply NMF with a common spatial prior across individuals (allowing us to account for individual variability in functional neuroanatomy in a standardized space), future work may investigate whether the varying degrees of stretching/squeezing of different cortical surface regions has an effect on associations between functional topography and behavior. Finally, our analyses focused on characterizing the cortical surface topography of functional brain networks and thus did not include analyses of subcortical regions. Future studies may use precision functional brain mapping approaches in subcortical areas[76,77] to further our understanding of the role of the subcortex in cognitive development. Future work may also expand upon our findings to explore other cognitive domains (e.g., social perception) that may be supported by distinct patterns of PFN topography as well as socio-demographic and environmental factors (e.g., socio-economic resources and structural racism) that may shape the development of PFN topography.

Critically, it is known that cognitive impairments in adulthood are common across diverse psychiatric illnesses including mood[26,78] and anxiety[79–81] disorders and our current first-line pharmacological treatments fail to target these cognitive symptoms[82,83]. Given that the functional topography of networks implicated in cognitive impairments and psychiatric illness (e.g., the fronto-parietal network[26]) tend to have the highest inter-individual heterogeneity[16], studies of personalized networks may be essential in better understanding these symptoms. Longitudinal studies of neurocognitive developmental trajectories may therefore also provide a critical link between functional brain organization in childhood and psychiatric illness in adulthood, with the potential to identify individuals at risk for cognitive impairments prior to the onset of psychiatric illness and in advance of treatment attempts that are likely to fail. Moreover, this study investigated PFNs in 9–10 year old children prior to the transition to adolescence; these children will be followed longitudinally into adulthood as part of the ABCD Study®. These results therefore lay a strong foundation for future work to uncover how PFNs derived at baseline may predict trajectories of change in cognitive functioning during development as longitudinal data is collected. Such studies may reveal distinct or overlapping neurobiological features that are predictive of future cognitive abilities and whose development tracks with the protracted development of higher-order cognition through childhood and adolescence.

Together, the findings of this study represent an important advance in our understanding of the link between individual differences in functional brain network organization and individual differences in cognitive functioning in youth. Further, these results successfully replicated prior findings[16] across two large samples of youth, providing compelling evidence that these observations are generalizable to new samples. Individual differences in cognition in youth are associated with critical physical, mental, social, and educational outcomes in adolescence and adulthood, ranging from academic achievement and financial success to psychopathology, risk-taking behaviors, and cardiovascular disease. Thus, our findings may inform studies that seek to develop interventions that could promote healthy neurocognitive development. By identifying PFNs whose functional topography is associated with cognition, we provide a foundation for future longitudinal studies of neurocognitive development and psychopathology.

## Methods
### Participants
Data were drawn from the ABCD study[36] baseline sample from the ABCD BIDS Community Collection (ABCC, ABCD-3165[37]), which included $n = 11,878$ children aged 9–10 years old and their parents/guardians collected across 21 sites. Parents and guardians provided written informed consent as part of the ABCD study. Institutional Review Board (IRB) approval was received from the University of California, San Diego and the respective IRBs of each study site. Inclusion criteria for this study included being within the desired age range (9–10 years old), English language proficiency in the children, and having the ability to provide informed consent (parent) and assent (child). Exclusion criteria included the presence of severe sensory, intellectual, medical or neurological issues that would have impacted the child's ability to comply with the study protocol, as well as MRI scanner contraindications. As depicted in Supplementary Fig. 3, we additionally excluded participants with incomplete data or excessive head motion, yielding a final sample of $n = 6972$.

To test the generalizability of our results, we repeated each of our analyses in both a discovery sample ($n = 3525$) and a separate replication sample ($n = 3447$) that were matched across multiple socio-demographic variables including age, sex, site, ethnicity, parent education, combined family income, and others[37,38]. Socio-demographic characteristics of participants in the discovery and replication samples may be found in Table 2. We observed no statistically significant differences between participants in the discovery and replication samples across any socio-demographic variables, psychotropic medication use (assessed by the Medication Inventory from the PhenX instrument and coded as in ref. 84), externalizing, internalizing, or problem behavior scores (assessed by the Child Behavior Checklist[85]), nor any of the three cognitive domains.

### Cognitive assessment
Participants completed a battery of cognitive assessments, including seven tasks from the NIH Toolbox (Picture Vocabulary, Flanker Test, List Sort Working Memory Task, Dimensional Change Card Sort Task, Pattern Comparison Processing Speed Task, Picture Sequence Memory Task, and the Oral Reading Test)[86] as well as two additional tasks (the Little Man Task and the Rey Auditory Verbal Learning Task)[87]. To reduce the dimensionality of these measures and focus our analyses on cognitive domains that explained the majority of behavioral variance in these tasks, we used scores in three previously-established cognitive domains derived from a prior study in this same dataset[40]: (1) general cognition, (2) executive function, and (3) learning/memory. In this study, a three-factor BPPCA model was applied to the aforementioned battery of nine cognitive tasks. Scores generated by varimax rotated loadings for this three-factor model for general cognition (highest loadings: Oral Reading Test, Picture Vocabulary, and Little Man Task), executive function (highest loadings: Pattern Comparison Processing Speed Task, Flanker Test, and Dimensional Change Card Sort Task), and learning/memory (highest loadings: Picture Sequence Memory

**Table 2 | Demographic characteristics and variables of interest in the matched discovery (n = 3525) and replication (n = 3447) samples**

| | Total | Discovery | Replication | P-value |
|---|---|---|---|---|
| Age (Months) | 119.5 (±7.5) | 119.5 (±7.6) | 119.5 (±7.5) | 0.94 |
| Sex (F) | 3494 (50.1%) | 1806 (51.2%) | 1688 (49.0%) | 0.06 |
| General Cognition | 0.1 (±0.7) | 0.1 (±0.7) | 0.1 (±0.7) | 0.18 |
| Executive Function | 0.0 (±0.8) | 0.0 (±0.8) | 0.1 (±0.7) | 0.23 |
| Learning/Memory | 0.1 (±0.7) | 0.1 (±0.7) | 0.0 (±0.7) | 0.22 |
| ADHD Medication | 536 (7.7%) | 283 (8.0%) | 253 (7.3%) | 0.30 |
| Antidepressant Medication | 122 (1.8%) | 69 (2.0%) | 53 (1.5%) | 0.20 |
| Antipsychotic Medication | 40 (0.6%) | 23 (0.7%) | 17 (0.5%) | 0.43 |
| CBCL Ext. | | | | 0.93 |
| Mean (SD) | 4.2 (±5.5) | 4.2 (±5.6) | 4.1 (±5.4) | |
| Missing | 1 (0.0%) | 0 (0.0%) | 1 (0.0%) | |
| CBCL Int. | | | | 0.50 |
| Mean (SD) | 5.0 (±5.5) | 5.0 (±5.6) | 4.9 (±5.4) | |
| Missing | 1 (0.0%) | 0 (0.0%) | 1 (0.0%) | |
| CBCL Prob. | | | | 0.99 |
| Mean (SD) | 17.3 (±17.2) | 17.4 (±17.5) | 17.1 (±16.9) | |
| Missing | 1 (0.0%) | 0 (0.0%) | 1 (0.0%) | |
| Household Income | | | | 0.28 |
| [<50 K] | 1645 (23.6%) | 806 (22.9%) | 839 (24.4%) | |
| [≥50 K & <100 K] | 1902 (27.3%) | 983 (27.9%) | 919 (26.7%) | |
| [≥100 K] | 2870 (41.2%) | 1451 (41.2%) | 1419 (41.2%) | |
| Missing | 552 (7.9%) | 284 (8.1%) | 268 (7.8%) | |
| Race | | | | 0.46 |
| White | 4723 (67.8%) | 2426 (68.8%) | 2297 (66.7%) | |
| Black | 910 (13.1%) | 442 (12.5%) | 468 (13.6%) | |
| Asian | 145 (2.1%) | 71 (2.0%) | 74 (2.1%) | |
| AIAN/NHPI | 41 (0.6%) | 23 (0.7%) | 18 (0.5%) | |
| Other | 242 (3.5%) | 114 (3.2%) | 128 (3.7%) | |
| Mixed | 823 (11.8%) | 410 (11.6%) | 413 (12.0%) | |
| Missing | 85 (1.2%) | 38 (1.1%) | 47 (1.4%) | |
| Parent Education | | | | 0.31 |
| <HS Diploma | 236 (3.4%) | 117 (3.3%) | 119 (3.5%) | |
| HS Diploma/GED | 542 (7.8%) | 254 (7.2%) | 288 (8.4%) | |
| Some College | 1781 (25.6%) | 905 (25.7%) | 876 (25.4%) | |
| Bachelor | 1885 (27.0%) | 980 (27.8%) | 905 (26.3%) | |
| Post Graduate Degree | 2519 (36.1%) | 1264 (35.9%) | 1255 (36.4%) | |
| Missing | 6 (0.1%) | 4 (0.1%) | 2 (0.1%) | |

There were no statistically significant differences between the discovery and replication samples in any of the attributes included in this table, evaluated using analyses of variance (ANOVAs), Fisher's exact tests, or standard Chi Squares analyses as appropriate.
*AIAN* American Indian/Alaska Native, *NHPI* Native Hawaiian and other Pacific Islander, *HS* high school, *GED* General Educational Development, *CBCL* child behavior checklist.

Task, Rey Auditory and Verbal Learning Task, and List Sort Working Memory Task) were downloaded directly from the ABCD Data Exploration and Analysis Portal.

**Image processing**
Imaging acquisition for the ABCD Study® has been described elsewhere[88]. As previously described[37], the ABCC Collection 3165 from which we drew our data was processed according to the ABCD-BIDS pipeline. This pipeline includes distortion correction and alignment, denoising with Advanced Normalization Tools (ANTS[89]), FreeSurfer[90] segmentation, surface registration, and volume registration using FSL

FLIRT rigid-body transformation[91,92]. Processing was done according to the DCAN BOLD Processing (DBP) pipeline which included the following steps: (1) de-meaning and de-trending of all fMRI data with respect to time; (2) denoising using a general linear model with regressors for signal and movement; (3) bandpass filtering between 0.008 and 0.09 Hz using a 2nd order Butterworth filter; (4) applying the DBP respiratory motion filter (18.582–25.726 breaths per minute), and (5) applying DBP motion censoring (frames exceeding an FD threshold of 0.2 mm or failing to pass outlier detection at +/− 3 standard deviations were discarded). Following preprocessing, we concatenated the time series data for both resting-state scans and three task-based scans (Monetary Incentive Delay Task, Stop-Signal Task, and Emotional N-Back Task) as in prior work[16] to maximize the available data for our analyses, though we note that all of our main results hold when we derive PFNs using only resting-state data from a subset of participants (n = 5968) who met our minimum threshold for clean resting-state data (Supplementary Table 4). We note that the original study[16] utilized a concatenated time series from both resting-state and two task-based scans yielding a time series length of 27 min 45 s prior to preprocessing, while the current study uses up to four resting-state scans and three task-based scans yielding a maximum time series length of 29 min 36 s after motion scrubbing. Participants with fewer than 600 remaining TRs after motion censoring or who failed to pass ABCD quality control for their T1 or resting-state fMRI scan were excluded. Since each of the participants has a variable number of frames remaining per run following our strict motion correction, it is not feasible to control for head motion independently for each run without biasing our results. We therefore use only the concatenated time series across all runs and the overall mean fractional displacement as a motion covariate in our analyses, so as not to inadvertently over- or under-correct for head motion for certain runs. We additionally excluded participants with incomplete data for our analyses (Supplementary Fig. 3). We then applied ComBat harmonization[58,59] using the *neuroCombat* package protecting age, family and sex as covariates, separately in the discovery and replication samples to harmonize the data across collection sites. Note that for our ridge regression models (described below), we chose to include data collection site as a covariate rather than apply ComBat harmonization to avoid leakage across our samples.

**Regularized non-negative matrix factorization**
As previously described[16,25], we used NMF[39] to derive individualized functional networks. NMF identifies networks by positively weighting connectivity patterns that covary, leading to a highly specific and reproducible parts-based representation[39,93]. Our approach was enhanced by a group consensus regularization term derived from previous work in an independent dataset[16] that preserves the inter-individual correspondence, as well as a data locality regularization term that makes the decomposition more robust to imaging noise, improves spatial smoothness, and enhances functional coherence of the subject-specific functional networks (see ref. 25 for details of the method; see also: https://github.com/hmlicas/Collaborative_Brain_Decomposition). As NMF requires nonnegative input, we re-scaled the data by shifting time courses of each vertex linearly to ensure all values were positive[25]. As in prior work, to avoid features in greater numeric ranges dominating those in smaller numeric range, we further normalized the time course by its maximum value so that all the time points have values in the range of [0, 1]. For this study, we used identical parameter settings as in prior validation studies[25], with the exception of an increase in the data locality regularization term from 10 to 300 to account for smaller vertices in *fslr* compared with *fsaverage5*.

**Defining individualized networks**
To facilitate group-level interpretations of individually-defined PFNs, we used a group consensus atlas from a previously published study in

an independent dataset[16] as an initialization for individualized network definition. In this way, we also ensured spatial correspondence across all subjects. This strategy has also been applied in other methods for individualized network definition[23,94]. Details regarding the derivation of this group consensus atlas can be found in previous work[16]. Briefly, group-level decomposition was performed multiple times on a subset of randomly selected participants and the resulting decomposition results were fused to obtain one robust initialization that is highly reproducible. Next, inter-network similarity was calculated and normalized-cuts[95] based spectral clustering method was applied to group the PFNs into 17 clusters. For each cluster, the PFN with the highest overall similarity with all other PFNs within the same cluster was selected as the most representative. The resulting group-level network loading matrix $V$ was transformed from *fsaverage5* space to *fslr* space using Connectome Workbench[96], and thus the resultant matrix had 17 rows and 59,412 columns. Each row of this matrix represents a functional network, while each column represents the loadings of a given cortical vertex.

Using the previously-derived group consensus atlas[16] as a prior to ensure inter-individual correspondence, we derived each individual's specific network atlas using NMF based on the acquired group networks ($17 \times 59,412$ loading matrix) as initialization and each individual's specific fMRI times series. See ref. 25 for optimization details. This procedure yielded a loading matrix $V$ ($17 \times 59,412$ matrix) for each participant, where each row is a PFN, each column is a vertex, and the value quantifies the extent each vertex belongs to each network. This probabilistic (soft) definition can be converted into discrete (hard) network definitions for display and comparison with other methods[19,23,94] by labeling each vertex according to its highest loading. Split-half reliability of the PFN loadings were assessed in ten participants who had the longest duration of low-motion quality resting-state data exceeding 20 min allowing us to derive PFNs in two 10-minute segments, as previously described in prior work[97], given the necessity of sufficient scan duration for the derivation of precision functional networks[20,21]. This analysis revealed high intraclass correlation coefficients for PFN loadings across all 17 networks (ICCs: 0.84–0.99) indicating excellent reliability of this measure (Supplementary Fig. 4) that aligns with what has been found in prior work[20,21]. For our univariate analyses, we calculated the total cortical representation of each PFN as the sum of network loadings across all vertices as previously[16], using the soft parcellation to account for spatial overlap across functional brain networks. As described in prior work[16], this measure of total cortical representation quantifies the spatial extent of each PFN on the cortical surface.

### Calculation of sensorimotor-association Axis Rank
To compute S-A axis rank for each PFN independently, we computed the average S-A rank across vertices for each PFN according to the hard network parcellation. Original S-A axis ranks by vertex represent the average cortical hierarchy across multiple brain maps[16] and were derived from https://github.com/PennLINC/S-A_ArchetypalAxis.

### Statistical analyses
Data analysis was performed using Matlab (R2022a), R (4.1.3) and Python (3.9). All code is available at https://github.com/PennLINC/keller-networks. For each statistical test reported, the data met the assumptions of the statistical tests used.

### Linear mixed-effects models
We used linear mixed effects models (implemented with the "lme4" package in *R*) to assess associations between PFN topography (total cortical representation) and performance in each cognitive domain while accounting for both fixed and random predictors. We note that these univariate association analyses are distinct from our multivariate predictive modeling approach (using ridge regression, as described below) in that we use a single measure (total cortical representation) per network rather than the multivariate pattern of PFN topography across all vertices (i.e., thousands of features per network). All linear mixed effects models included fixed effects parameters for age, biological sex (self-reported sex assigned at birth), head motion (mean fractional displacement), as well as a random intercept for family (accounting for siblings). Note that data were harmonized across sites prior to linear mixed-effects model analyses, and thus there was no need to include a random intercept for site.

### Ridge regression
We trained ridge regression models to predict cognitive performance in each of the three cognitive domains (general cognition, executive function, and learning/memory) using the functional topography (vertex-wise network loading matrices) of each participant's PFNs. In line with the recommendation that predictive models of brain-behavior associations be trained on multivariate patterns rather than univariate measures[98], these predictive models were trained on concatenated network loading matrices across the 17 PFNs. Independent network models were also trained on the network-wise loadings at each vertex. All models included covariates for age, sex, site, and motion (mean FD).

Our primary ridge regression models were trained and tested on the ABCD reproducible matched samples[37,38] using nested two-fold cross-validation (2F-CV), with outer 2F-CV estimating the generalizability of the model and the inner 2F-CV determining the optimal tuning parameter ($\lambda$) for the ridge regression model. For the inner 2F-CV, one subset was selected to train the model under a given $\lambda$ value in the range $[2^{10}, 2^9, ..., 2^4, 2^5]$ (i.e., 16 values in total)[16], and the remaining subset was used to test the model. This procedure was repeated 2 times such that each subset was used once as the testing dataset, resulting in two inner 2F-CV loops in total. For each $\lambda$ value, the correlation $r$ between the actual and predicted outcome as well as the mean absolute error (MAE) were calculated for each inner 2F-CV loop, and then averaged across the two inner loops. The sum of the mean correlation $r$ and reciprocal of the mean MAE was defined as the inner prediction accuracy, and the $\lambda$ with the highest inner prediction accuracy was chosen as the optimal $\lambda$[16]. Of note, the mean correlation $r$ and the reciprocal of the mean MAE cannot be summed directly, because the scales of the raw values of these two measures are quite different. Therefore, we normalized the mean correlation $r$ and the reciprocal of the mean MAE across all values and then summed the resultant normalized values.

To ensure that our matched discovery and replication sample selection procedure did not bias our results, we performed repeated random cross-validation over 100 iterations, each time randomly splitting the sample and repeating the nested 2F-CV procedure to generate a distribution of prediction accuracies for each model. Furthermore, we used permutation testing to generate null distributions for both the primary models and the repeated random cross-validation models by randomly shuffling the outcome variable. Supplementary Fig. 1 depicts the sum of model weights by PFN for the primary ridge regression models in each of the matched samples. To ensure that our results were not overfit as a result of leakage across samples by the cognitive outcome variables derived in the whole sample[40], we also trained ridge regression models to predict cognitive scores derived by two independent principal components analyses in the discovery and replication samples separately. Repeating our main analyses with these new predictive models, we find functionally identical results as shown in Supplementary Fig. 5. We also found consistent prediction accuracy results when training our ridge regression models to predict performance in each cognitive task independently (Supplementary Table 5). We also tested whether the prediction accuracy from these ridge regression models was associated with the total size of each PFN, defined as the number of vertices belonging to each PFN in the hard

parcellation (Supplementary Fig. 6). However, we note that association networks ranked higher along the S-A axis tend to occupy a larger portion of the cortical surface than sensorimotor networks ranked lower along the axis (see following section on associations between prediction accuracy and S-A axis rank). We further note that our multivariate ridge-regression analyses leverage the full richness and complexity of PFN topography across vertices, while our univariate linear mixed effects analyses of total cortical representation use a single scalar summary statistic (total cortical representation) per PFN. Thus, effect sizes for these multivariate prediction analyses are expected to be larger than effect sizes for our univariate association analyses.

### Associations between prediction accuracy and S-A axis rank
To compute associations between the prediction accuracy of each individual network model and the average S-A rank for each network, we used Spearman's rank correlations for each of the three cognitive domains: general cognition, executive function, and learning/memory. To determine whether the alignment between these two spatial maps were driven specifically by the S-A axis, we used spatial permutation testing[56] (Spin Tests; https://github.com/spin-test/spin-test). The spin test is a spatial permutation method based on angular permutations of spherical projections at the cortical surface. Critically, the spin test preserves the spatial covariance structure of the data, providing a more conservative and realistic null distribution than randomly shuffling locations. With this approach, we applied 1000 random rotations to spherical representations of S-A axis rank across the cortical surface, each time re-computing the average S-A rank within each PFN and calculating Spearman correlations between prediction accuracy and the permuted average S-A rank in each PFN to generate a null distribution. We then compared the true Spearman correlation value to the null distribution of spatially permuted Spearman correlations by rank ordering.

### Reporting summary
Further information on research design is available in the Nature Portfolio Reporting Summary linked to this article.

## Data availability
Data used in the preparation of this article were obtained from the Adolescent Brain Cognitive Development Study® (https://abcdstudy.org), held in the NIMH Data Archive (NDA). Only researchers with an approved NDA Data Use Certification (DUC) may obtain ABCD Study data.

## Code availability
Code is available at https://github.com/PennLINC/keller-networks[99].

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

## Acknowledgements

This study was supported by grants from the National Institutes of Health: R01MH113550 (T.D.S. and D.S.B.), R01MH120482 (T.D.S.), R01EB022573 (T.D.S.), R37MH125829 (T.D.S. and D.A.F.), K99MH127293 (B.L.), F31MH123063-01A1 (A.R.Pi.), R01MH123550 (R.T.S.), R01MH112847 (R.T.S.), R01MH123563 (R.T.S.). A.S.K. was supported by a Neuroengineering and Medicine T32 Fellowship from the National Institute of Neurological Disorders and Stroke (5T32NS091006-08) and a Neurodevelopment and Psychosis T32 Fellowship from the National Institute of Mental Health (5T32MH019112-32). A.R.Pi. was supported by the Stanford School of Medicine Dean's Fellowship. S.S. was supported by R25MH119043, T32NS091008, and a 2022 Young Investigator Grant from the Brain & Behavior Research Foundation. Additional support was provided by the Penn-CHOP Lifespan Brain Institute. Data used in the preparation of this article were obtained from the Adolescent Brain Cognitive Development℠ (ABCD) Study (https://abcdstudy.org), held in the NIMH Data Archive (NDA). This is a multisite, longitudinal study designed to recruit more than 10,000 children age 9–10 and follow them over 10 years into early adulthood. The ABCD Study® is supported by the National Institutes of Health and additional federal partners under award numbers U01DA041048, U01DA050989, U01DA051016, U01DA041022, U01DA051018, U01DA051037, U01DA050987, U01DA041174, U01DA041106, U01DA041117, U01DA041028, U01DA041134, U01DA050988, U01DA051039, U01DA041156, U01DA041025, U01DA041120, U01DA051038, U01DA041148, U01DA041093, U01DA041089, U24DA041123, U24DA041147. A full list of supporters is available at https://abcdstudy.org/federal-partners.html. A listing of participating sites and a complete listing of the study investigators can be found at https://abcdstudy.org/consortium_members/. ABCD consortium investigators designed and implemented the study and/or provided data but did not necessarily participate in the analysis or writing of this report. This manuscript reflects the views of the authors and may not reflect the opinions or views of the NIH or ABCD consortium investigators. The ABCD data repository grows and changes over time. The ABCD data used in this report came from [NIMH Data Archive Digital Object Identifier 10.15154/1523041]. DOIs can be found at https://nda.nih.gov/abcd.

## Author contributions

A.S.K. analyzed the data and wrote the initial draft of the manuscript. A.R.Pi. and S.S. replicated all analyses. A.R.Pi., Z.C., M.A.B., N.B., G.M.C., E.F., T.J.H., A.H., H.L., O.M.-D., A.Pe. and Y.F. contributed to data pre-processing. A.C. and R.T.S. contributed to data harmonization. A.S. contributed to figure and table generation and formatting. V.J.S., R.B., A.F.A.-B., C.D., B.L. and D.R.R. contributed scientific guidance. D.A.F. and T.D.S. jointly supervised the project. All authors A.S.K., A.R.Pi., V.J.S., Z.C., M.A.B., R.B., A.F.A.-B., N.B., A.C., G.M.C., C.D., E.F., T.J.H., A.H., B.L., H.L., O.M.-D., D.R.R., A.Pe., S.S., A.S., R.T.S., Y.F., D.A.F. and T.D.S. reviewed and provided feedback on the final draft of the manuscript.

## Competing interests

R.B. reports owning stock in Taliaz Health and serving on the scientific boards of Taliaz Health and Zynerba Pharmaceuticals outside the submitted work. Dr. Shinohara has consulting income from Genentech/Roche and Octave Bioscience. All other authors report no competing interests.

## Additional information

[1]Penn Lifespan Informatics and Neuroimaging Center, University of Pennsylvania, Philadelphia, PA 19104, USA. [2]Department of Psychiatry, University of Pennsylvania, Philadelphia, PA 19104, USA. [3]Chinese Institute for Brain Research, Beijing, China. [4]Department of Child and Adolescent Psychiatry and Behavioral Science, Children's Hospital of Philadelphia, Philadelphia, PA 19104, USA. [5]Lifespan Brain Institute, Children's Hospital of Philadelphia and Perelman School of Medicine, University of Pennsylvania, Philadelphia, PA 19104, USA. [6]Masonic Institute for the Developing Brain, Institute of Child Development, College of Education and Human Development, Department of Pediatrics, Medical School, University of Minnesota, Minneapolis, MN 55414, USA. [7]Penn Statistics in Imaging and Visualization Center, Department of Biostatistics and Epidemiology, University of Pennsylvania, Philadelphia, PA 19104, USA. [8]Department of Radiology, University of Pennsylvania, Philadelphia, PA 19104, USA. [9]University of Minnesota Informatics Institute, University of Minnesota, Minneapolis, MN 55414, USA. [10]These authors jointly supervised this work: Damien A. Fair, Theodore D. Satterthwaite. ✉e-mail: sattertt@pennmedicine.upenn.edu

