## [Peer Review File · Nature Communications]

Personalized Functional Brain Network Topography is
Associated with Individual Differences in Youth CognitionREVIEWER COMMENTS

Reviewer #1 (Remarks to the Author):

In the paper, "Personalized Functional Brain Network Topography Predicts Individual Differences in Youth Cognition", Keller and colleagues use a large sample (n=6,972) of resting-state and task data from the ABCD Study dataset to delineate individualized functional networks to relate to cognitive ability. Methodologically, this paper is as strong as it gets, with matched discovery and replication samples allowing for replication of all results. The paper was also very well written. Conceptually, I am not sure of that there is a real advance from the previous paper by this group (Cui et al., 2020). The authors found similar prediction accuracy in ABCD (a larger dataset than use in Cui et al [PNC]), but aside from that I'm not sure what new conceptual advance this paper makes. However, I do applaud replication attempts, bolstering confidence in the generalizability of the results. In addition to limited conceptual advances, I have several major concerns that I will detail below.

1. The major concern I have with the paper in its current form is the lack of any mention of socioeconomic status/resources (SES/SER), given the correlation between cognitive ability and SES. Relative to most cognitive neuroscience datasets, ABCD has several measures related to SES, including the Area Deprivation Index. Without any correction for SES, how can the authors be confident in their claims regarding cognition prediction accuracy?
2. Group vs. individual level networks. I applaud the author's effort in this regard. As the authors note, given the recent brain behavior paper by Marek et al (2022), one suggestion for improving power is the use of individualized networks rather than group networks. I found the approach methodologically sound. However, the paper could be strengthened by directly comparing group vs. individual level prediction accuracy for these phenotypes. Otherwise, there is no baseline to anchor justification for individual approaches.
3. Inference based off of individual level data in ABCD. I think it's great that the authors included both task and resting state ABCD data to boost the amount of data going into their models. That said, just by doing this, these data cannot be assumed to be individually precise. Given the fact that the reliability of the resting state correlation data puts a ceiling on the reliability of the individualized networks (i.e., reliability of networks [once removed from correlation data] cannot be higher than reliability of functional connectivity itself), I think it prudent to show both the reliability of the concatenated rest + task time series and the reliability of the individualized networks. To be convinced of their reliability, I think the authors must quantify that first before relating to non-brain phenotypes. I think this is very important given the prediction accuracies are similar to Cui et al 2020, in which substantially less data per person was used.
4. Prediction accuracy of executive function and learning and memory. Given the NIH Toolbox data were broken into components using PCA, shouldn't we expect, by definition, that the first component would yield the highest prediction accuracy since it contains the greatest amount of between-person variance? To be clear, I am not saying this is not a valid or reproducible result; I am not sure what to make of the fact that these prediction accuracies are lower. It seems to just be epiphenomenal.
5. Similarly, the S-A axis results (Figure 5) also seem epiphenomenal. Since FPN shows a positive association with general cognitive ability and motor networks a negative relationship, does not a positive association along the S-A axis have to exist?

Reviewer #2 (Remarks to the Author):

This manuscript investigates individual differences in cognition during childhood and attempts to demonstrate that personalized functional networks (PFNs) are associated with general cognition. The associations are then claimed to be validated using a ridge regression predictive model of cognition and the authors also show that prediction accuracy increases along the cortex's sensorimotor-

association organizational atlas.

Strength of this work is that it replicates results published by this same group on slightly different data using slightly different measures, published in Neuron in 2020. Another strength is the use of personalized functional networks as input to both the association studies and as input to the predictive models.

1. Enthusiasm is reduced by the weak associations shown in figures 2 (b,c,d, f, g, h), and figure 4, yet much of the discussion and conclusions are focused on these associations.

2. The predictive models predict factors other than what is shown in the association studies and it's not clear why. For example this reviewer suspects it would be difficult to build a predictive model that works based on the associations shown in figure 2 (b,c,d) and (f,g,h) so then why show these associations? (e.g. total representation is not used in the predictive models)

3. The primary conclusions are focused on the associations with an acknowledgement that such association studies are not widely replicable (with reference to the Marek 2022 Nature paper) but no mention is made of the paper by Rosenberg and Finn (Nature Neuro 25(7), 2022) where they specifically point out that associations can be believed if one can build predictive models from the data. Yet that is not what is done here. If the authors want to discuss associations between total representation and cognitive variables, then they should take those associations and build predictive models from them – yet this is not what is done.

The Marek et al argument of poor replicability can be found with weak associations even with large N, hence the need for predictive models on the direct associations – if they're not predictive, the associations are not meaningful ala the Rosenberg and Finn paper.

4. The predictive models provided have outstanding performance, but it is unclear why, given the weak associations described. Many steps are taken between the associations shown however and the final predictive models that are built.

a) How exactly was motion dealt with. Figure 2 shows results from a mixed effect analysis but it appears that motion played a significant role in this analysis. There should be more clarity on how age, sex, motion, and site were controlled for in this analysis.

b) It appears to be the case that probabilistic PCA was performed on the cognitive measures and then predictive models built using the 1st, 2nd and 3rd PCs. If the PCA is performed on the entire sample before it was split into the discovery and replication set then the data is contaminated and this could account for the high prediction performance. Lines 471-474 seem to imply this was run on all the subjects together.

c) Probably the best way to resolve this disconnect with the predictive models and the associations is to show the associations for the factors that were predicted or actually build the predictive model based on the associations under discussion.

5. Figure 3a and Figure 4a&e seem to fail the eye-ball test in that the line running through the data does not appear to follow the data. How are the lines derived and why don't they match the data? Was there a constraint placed on the regression to make it go through zero? (such a constraint would not really be valid here)

Reviewer #3 (Remarks to the Author):

Keller & co-authors investigate how functional connectivity topography relate to cognitive and behavioral characteristics in a large sample of children (ABCD sample). Functional topography was

characterized using a non-negative matrix factorization technique, to define 17 overlapping networks on each subject's cortical surface. Importantly, this technique allows for networks to be derived in a data-driven manner for each subject's functional data. The loadings for each of the 17 networks were summed and associated with a general cognitive trait, replicating a previously conducted analyses (although, with small effect sizes). Notably, this replication has been conducted in a much larger dataset than the previous study. Another replication was conducted using the network loadings, both combined and separated out, to predict cognition, executive function, and learning/memory. These analyses demonstrate the importance of data belonging to the fronto-parietal networks for these predictions. Finally, prediction accuracy was shown to rank correlate with the so-called S-A axis, placing these results within the larger context of studies of cortical gradients. The manuscript does well to highlight its contributions, namely that this investigation is a replication of previous work, with a larger sample and different preprocessing pipeline, and that the NMF approach allows for individualized representations of functional networks to be estimated cortex wide.

Main Critiques:

The results shown in Figure 2, use "total cortical representation" as the brain feature of interest. I find that more explanation, and possible exploration, of this feature should be provided for readers, to understand the initial result of this manuscript. Even in citation 16, it was not clear what this feature means. As I understand it, it is a sum of all NMF weights (for a specific network) across the whole cortex. I think readers might benefit from speculation of why greater the sum of these weights would relate positively to increased general cognition capacity. I am sympathetic to the idea that we don't know what it means at the biological level – and in fact, your study might provide evidence to suggest an interpretation of it: that greater representation is meaningful for prediction. To be clear, I think the authors provide excellent discussion of why the topography of fronto-parietal networks are important to predict these individual differences (the section, Association network topography supports domain-general cognitive abilities, is great). I'd like to see additional explanation that focuses on what "total cortical representation" is or what it conceptually represents (which could be additionally included in methods, even), since it is central to the manuscript.

Relatedly, I could imagine that there might be some sources of potential bias, related to head size for the total cortical representation feature. Individual's surfaces need to be stretched and squeezed to the common fsLR surface. What I am getting at is the possibility that cortical representation of a specific brain network could be influenced by this transformation from individual to common surface space. There is the possibility that a hypothetical functional area that initially occupied little cortical surface in subject space, has the potential to be expanded, if that area needs to be stretched, and thus, the functional data interpolated to occupy a larger space to match fsLR. I am not sure the NMF clustering on this data would mitigate this necessarily. Have you investigated ways to modulate the loading variables by some stretching/squeezing feature? Akin to how in Voxel-Based Morphometry, grey matter probability values are scaled by Jacobian determinants to account for local compression and stretching after the grey matter probability values that have been warped to the common space. If this computational undertaking is unfeasible for the authors, (and the authors agree with the logic of my critique) I imagine this could be noted as a limitation. This critique only affects mainly "total cortical representation" feature. Although, I do wonder how Prediction Accuracy is related to the original "size" (but of course, these networks actually have loadings across the cortex) of each network map (referring to the visualization in Figure 1e); just eyeballing the results, it looks like the least predictive networks (4, 13, 15, and 6) cover much less space than the most predictive ones (3, 9, and 12). It might be useful to plot the prediction accuracy versus network "size" (however you might define that—you might need to pick a threshold on the weights) and include that as a supplementary figure.

I understand (and appreciate) that the authors used state-of-the-art processing pipelines, have already eliminated thousands of subjects based on motion, and have censored the time series.

However, given that meanFD is found to be a significant term in the mixed effects model for each of the PFN networks, in both the replication and discovery cohorts (Table 1 results), I'll provide one more suggestion. Since data is combined from multiple runs (rest and three task scans), is there an effect of motion that could be localized to each run. In other words, could # censored frames, for example, for each run be included as 4 separate nuisance covariates in the model. This might account for differences in participant vigilance (or restlessness) in the context of rest versus the different tasks, that could interact with the associations to the cognitive and behavioral variables.

Minor comments:

For Figure 2, it would be nice to make the axes have the same range in (a) and (e), to make comparisons easier on the eye.

In the limitations it was noted that data from two (but I thought it was three, as noted in the methods...) were task regressed, but the methods do not detail how the task regression was performed.

Reviewer #4 (Remarks to the Author):

Thank you for the opportunity to review the manuscript Personalized Functional Brain Network Topography Predicts Individual Differences in Youth Cognition. The study is well-written, interesting, and timely. Statistical methodology is robust and leverages a large-scale dataset from the ABCD study. The study examines inter-individual heterogeneity in functional brain network organization using functional mapping and cross-validated models. The authors report three frontoparietal PFNs that were significantly associated with general cognition across both discovery and replication samples, with a somatomotor network inversely associated with cognitive performance. The study explores a much needed area of research in functional connectivity, inter-individual heterogeneity in functional network topography, and brain-wide association studies. However, it is largely a replication of Cui et al., (2020) with the ABCD dataset. Nonetheless, the study is elegant and important to the field.

1. The authors concatenated across resting-state and task fMRI. This approach is understandable to provide a sufficient voxel timecourse for connectivity analyses. However, I have two concerns. First, nuanced detail is lost regarding potential differences in resting-state vs task-based fMRI in predicting general cognition or subdomains of cognition. Second, the three tasks are related but distinct and capture different aspects of cognition and emotion processing. There is an opportunity here to provide more nuanced detail of PFNs, that builds on but also extends the initial study from Cui et al. (2020). It would be helpful if the authors conducted additional follow up, sub-analyses for resting-state vs task to examine associations between PFNs and cognition. In addition, to fully test replication with Cui et al., which leveraged resting-state and tasks of emotion perception and n-back, I would like to see a sub-analysis repeated with resting-state and conceptually similar tasks such as n-back from ABCD.

2. One strength of the ABCD study for brain-wide association studies and fMRI is the sample heterogeneity. However, because the dataset includes youth with a range of psychiatric symptom severities, the number of children taking psychotropic medications in the discovery/replication samples should also be reported and accounted for in models. This is essential given the effect of psychotropic medications on functional connectivity and the heterogeneity of the ABCD sample.

3. Related to above, were there differences between the discovery/replication samples in terms of child psychopathology? The authors may want to consider testing for differences between the samples using the CBCL externalizing and internalizing scales, or the CBCL overall problem behavior scale. The authors conducted random cross-validation, which is great. However, additional follow-up analyses are essential for interpretation to demonstrate findings are not impacted by severity of psychopathology among discovery and replication samples.

4. The authors may also want to consider greater consideration for race/ethnicity in sub-analyses. An advantage of large-scale imaging datasets, such as the ABCD study, is the opportunity to explore differential effects of functional connectivity that may be influenced by race/ethnicity. Similarly, it would be helpful to test if there is an association between network topography and sex.

5. Why not test if functional topography predicts social perception? This would complement analyses to test for specificity of PFNs for cognition vs other psychological processes, as well as inform future work regarding PFNs and social perception.

6. The effect sizes here are good but not great. Considering the need for translation of neuroscience and neuroimaging findings that can inform clinical decision making in child and adolescence mental health, I would like to see this issue explored in greater detail in the discussion, particularly in the context of predictive modeling, comparisons of effect sizes to other imaging studies using ABCD or comparable large-scale datasets and/or methodological approaches, and considerations as well as challenges for clinical translational potential of small to medium effect sizes in brain-wide association studies and fMRI.

7. How was the tuning parameter for the ridge regression determined?

8. Why not test the generalization of predictive models in the current study to Cui et al. 2020 (and vice versa)?

9. The authors state that "nonsignificant differences between participants in the discovery and replication samples were present across any socio-demographic variables, nor were there any significant differences in scores across the three cognitive domains." Were there associations between socio-demographic variables and connectivity?

10. The authors state that the findings represent "...a critical step toward understanding healthy neurocognitive development." However, this study leveraged a dataset with a heterogeneous sample of youth with varying levels of internalizing/externalizing symptoms (with ~40% in the clinically significant range). These findings don't necessarily represent an entirely homogeneous sample of unaffected, healthy control youth or "healthy" neurocognitive development. The authors may want to consider rephrasing.

Minor comments:

11. In the introduction and sections of the discussion, the authors could consider expanding more on the importance of inter-individual heterogeneity in network organization and predictive models, particularly due to the role of the frontoparietal network across domains of psychopathology.

12. Please specify what the variable "family" refers to (e.g., income, siblings, environment). This seems overly broad and should be clarified.

13. Table 2: what was the proportion of girls vs boys in the total and discovery/replication samples? This is not specified.

REVIEWER #1

In the paper, “Personalized Functional Brain Network Topography Predicts Individual Differences in Youth Cognition”, Keller and colleagues use a large sample (n=6,972) of resting-state and task data from the ABCD Study dataset to delineate individualized functional networks to relate to cognitive ability. Methodologically, this paper is as strong as it gets, with matched discovery and replication samples allowing for replication of all results. The paper was also very well written. Conceptually, I am not sure of that there is a real advance from the previous paper by this group (Cui et al., 2020). The authors found similar prediction accuracy in ABCD (a larger dataset than use in Cui et al [PNC]), but aside from that I'm not sure what new conceptual advance this paper makes. However, I do applaud replication attempts, bolstering confidence in the generalizability of the results. In addition to limited conceptual advances, I have several major concerns that I will detail below.

1. The major concern I have with the paper in its current form is the lack of any mention of socioeconomic status/resources (SES/SER), given the correlation between cognitive ability and SES. Relative to most cognitive neuroscience datasets, ABCD has several measures related to SES, including the Area Deprivation Index. Without any correction for SES, how can the authors be confident in their claims regarding cognition prediction accuracy?

We appreciate the reviewer’s point about the importance of investigating the role of socio-economic status (SES) in studies of cognition and brain development. Prior studies using the ABCD dataset have rigorously explored associations among SES, cognitive functioning, and brain network properties (Ellwood-Lowe et al., 2021) and it has been consistently demonstrated that higher SES is associated with better performance on a wide range of cognitive tasks (Gonzalez et al., 2020; Kirlic et al., 2021; R. C. Thompson et al., 2022; Tomasi & Volkow, 2021). In line with the reviewer’s suggestion, we have conducted an additional analysis using Area Deprivation Index (ADI) as a measure of SES in our analyses of cognition prediction accuracy. Specifically, we trained two new sets of ridge regression models using the same procedure as in our main results to predict three domains of cognitive functioning (General Cognition, Executive Function, and Learning/Memory): the first set of models used only ADI as a predictor of cognitive functioning to reveal the baseline prediction accuracy that could be achieved, while the second set of models used SES in addition to the multivariate pattern of personalized functional brain network (PFN) topography (as in our previous results). As shown in the new **Supplementary Table 1** copied below, we now highlight the differences in prediction accuracies across ridge regression models including vs. not including SES.

Prediction Accuracy	Discovery		Replication	
	r	p	r	p
General Cognition				
SES	0.26	1.35 x 10 ⁻⁵¹	0.28	4.77 x 10 ⁻⁵⁸
PFN Topography	0.41	3.05 x 10 ⁻¹⁴⁶	0.45	3.85 x 10 ⁻¹⁷⁴
SES + PFN Topography	0.43	1.01 x 10 ⁻¹⁵¹	0.46	1.80 x 10 ⁻¹⁷¹
Executive Function				
SES	0.07	1.14 x 10 ⁻⁴	0.09	3.25 x 10 ⁻⁷
PFN Topography	0.17	1.37 x 10 ⁻²³	0.16	5.48 x 10 ⁻²²
SES + PFN Topography	0.17	7.18 x 10 ⁻²²	0.17	2.59 x 10 ⁻²³
Learning/Memory				
SES	0.13	2.96 x 10 ⁻¹³	0.16	4.46 x 10 ⁻¹⁹
PFN Topography	0.27	2.06 x 10 ⁻⁶¹	0.27	2.91 x 10 ⁻⁵⁷
SES + PFN Topography	0.27	3.53 x 10 ⁻⁵⁷	0.27	2.35 x 10 ⁻⁵⁶

Supplementary Table 1. Predictive models incorporating socio-economic status (SES).

Prediction accuracy, measured as the correlation r between actual and predicted cognitive performance, is shown for ridge regression models trained to predict cognitive performance across three domains (General Cognition, Executive Function, and Learning/Memory) across both discovery and replication samples. Results from three sets of predictive models are shown: “SES” refers to models trained only on socio-economic status as measured by the areal deprivation index; “PFN Topography” refers to models trained on the multivariate pattern of personalized functional brain network (PFN) topography for each individual (as presented in the main text and in Figures 2 and 3); and “SES + PFN Topography” refers to models trained on both socio-economic status and PFN topography. Although SES is a significant predictor of cognitive functioning, models trained on PFN topography yield much stronger predictions of cognitive performance than SES alone, and the addition of SES information to models trained on PFN topography does not substantially increase prediction accuracy. This observation suggests that the spatial topography of individually-defined functional brain networks accounts for additional inter-individual variance in cognitive performance beyond what is accounted for by SES alone.

The modest prediction accuracies for the SES-only models suggest that, in line with prior studies, SES is a significant predictor of cognitive functioning. Comparing prediction accuracies across models, we find that models trained on PFN topography yield much stronger predictions of cognitive performance than SES alone, and that the addition of SES information to models trained on PFN topography does not substantially increase

prediction accuracy. This observation suggests that SES is associated with some of the inter-individual heterogeneity in PFN topography but does not fully account for the association between PFN topography and cognitive performance. Put another way, the spatial topography of individually-defined functional brain networks accounts for additional inter-individual variance in cognitive performance beyond what is accounted for by SES alone. This finding is now described in the revised manuscript on lines 185-190:

“Given that many prior studies in this dataset have demonstrated associations between socio-economic status and cognitive functioning,⁴⁴⁻⁵³ we also note that our predictive models trained on PFN topography outperformed models trained on socio-economic status as measured by areal deprivation index (ADI) alone, and we observed little to no improvement in prediction accuracy when models were trained on both ADI and PFN topography together (**Supplementary Table 1**).”

To further demonstrate that the inclusion of ADI as a covariate does not alter our univariate association results, we conducted an additional sensitivity analysis now included in the new **Supplementary Table 3** copied below. This analysis confirmed that all of our univariate association results remained significant with the inclusion of this covariate, with general cognition still significantly associated with the total cortical representation of all three fronto-parietal PFNs across both the discovery and replication samples.

Predictors	PFN 3				PFN 15				PFN 17			
	β	Std. Error	t	p _{bonf}	β	Std. Error	t	p _{bonf}	β	Std. Error	t	p _{bonf}
Discovery												
Intercept	0.01	0.02	0.42	0.672	0.08	0.02	3.24	1.21 x 10⁻³	-0.04	0.02	-1.80	7.26e-02
Age	-0.04	0.02	-2.23	0.026	-0.03	0.02	-1.43	0.152	-0.05	0.02	-2.59	9.65 x 10⁻³
Sex	-0.05	0.04	-1.51	0.131	-0.16	0.03	-4.53	6.23 x 10⁻⁶	0.07	0.03	1.91	0.0557
Mean FD	0.12	0.02	7.07	1.83 x 10⁻¹²	0.12	0.02	6.65	3.36 x 10⁻¹¹	0.04	0.02	2.08	0.038
SES	-0.00	0.02	-0.14	0.892	0.01	0.02	0.51	0.610	0.02	0.02	1.34	0.179
General Cognition	0.08	0.03	3.20	1.40 x 10⁻³	0.08	0.03	3.10	1.96 x 10⁻³	0.11	0.03	4.06	5.09 x 10⁻⁵
Replication												
Intercept	0.01	0.03	0.44	0.661	0.07	0.03	2.62	8.86 x 10⁻³	-0.02	0.03	-0.78	0.436
Age	-0.02	0.02	-0.86	0.392	-0.06	0.02	-3.13	1.77 x 10⁻³	-0.05	0.02	-2.68	7.35 x 10⁻³
Sex	-0.05	0.04	-1.43	0.154	-0.15	0.04	-4.10	4.22 x 10⁻⁵	0.03	0.04	0.86	0.388
Mean FD	0.16	0.02	9.32	2.18 x 10⁻²⁰	0.09	0.02	4.84	1.37 x 10⁻⁶	0.04	0.02	2.41	0.016
SES	0.03	0.02	1.50	0.133	0.05	0.02	2.71	6.71 x 10⁻³	0.04	0.02	2.36	0.018
General Cognition	0.07	0.03	2.53	0.011	0.07	0.03	2.57	0.010	0.10	0.03	3.93	8.51 x 10⁻⁵

Supplementary Table 3. Sensitivity analyses controlling for socio-economic status (SES).

Linear mixed effects models associating general cognition with the total cortical representation of

fronto-parietal PFNs remain significant across both the discovery and replication samples when controlling for socio-economic status (SES) as measured by areal deprivation index.

Further, to encourage future research studies to explore this important line of investigation in more depth, we have also included the following text in our revised manuscript on lines 441-444:

“Future work may also expand upon our findings to explore other cognitive domains (e.g., social perception) that may be supported by distinct patterns of PFN topography as well as socio-demographic and environmental factors (e.g., socio-economic resources, race/ethnicity, and structural racism) that may shape the development of PFN topography.”

2. Group vs. individual level networks. I applaud the author’s effort in this regard. As the authors note, given the recent brain behavior paper by Marek et al (2022), one suggestion for improving power is the use of individualized networks rather than group networks. I found the approach methodologically sound. However, the paper could be strengthened by directly comparing group vs. individual level prediction accuracy for these phenotypes. Otherwise, there is no baseline to anchor justification for individual approaches.

While we agree that in general it can be interesting to compare measures (e.g., functional connectivity) derived from group atlases vs. individualized networks, our approach is aimed at understanding the *spatial arrangement* of networks on the cortical surface rather than *network-level measures* like functional connectivity. As such, it is not possible for us to perform our analyses using a group-level atlas as a comparison, since a group atlas would yield only one set of values (vertex-wise network loadings) that would by definition be the same across all individuals. Our approach therefore provides a new avenue by which to relate individual differences in functional brain network topography to individual differences in cognition that would not be possible by looking at group-averaged functional brain network topography alone. This novelty has been previously described by Bijsterbosch et al. (2021), who wrote that studying individualized networks “offers insights into previously untapped sources of between-subject variation such as differences in the size, shape, position and non-topological variation of brain areas and networks (Bijsterbosch et al., 2018; Glasser et al., 2016; Kong et al., 2019).” (Bijsterbosch et al., 2021)

3. Inference based off of individual level data in ABCD. I think it’s great that the authors included both task and resting state ABCD data to boost the amount of data going into their models. That said, just by doing this, these data cannot be assumed to be individually precise. Given the fact that the reliability of the resting state correlation data puts a ceiling on the reliability of the individualized networks (i.e., reliability of networks [once removed from correlation data] cannot be higher than reliability of functional connectivity itself), I think it prudent to show both the reliability of the concatenated rest + task time series and the reliability

of the individualized networks. To be convinced of their reliability, I think the authors must quantify that first before relating to non-brain phenotypes. I think this is very important given the prediction accuracies are similar to Cui et al 2020, in which substantially less data per person was used.

Prior literature on individually-defined functional brain networks has demonstrated that individual differences in the spatial organization of such networks are remarkably stable, with minimal day-to-day or task-state variability (Gratton et al., 2018). However, as suggested, we have included a new supplementary analysis in which we assess the split-half reliability of the PFN loadings. As in prior work (Hermosillo et al., 2022), we assessed split-half reliability in ten participants who had a sufficient duration of low-motion high-quality resting-state data exceeding 20 minutes. We first computed PFNs in each 10-minute half of the data using the approach described in the current manuscript and then calculated the split-half reliability for each participant and each PFN as the intraclass correlation coefficient (ICC) between the vertex-wise loadings in each half of the data. To contextualize our results: ICC scores greater than 0.9 are considered “excellent” while scores between 0.75 to 0.9 are considered “good”, scores between 0.5 and 0.75 are considered “moderate” and scores below 0.5 are considered “poor” (Koo and Li, 2017). Across all seventeen PFNs and all ten participants, all ICC scores fell within a range of 0.84 to 0.99, with the majority of ICC scores (96%) falling in the “excellent” category ($ICC > 0.9$). ICC scores for each participant and each network are plotted in the figures below, which we now include in the new **Supplementary Figure 4** copied below:

Supplementary Figure 4. Split-Half Reliability of PFN Topography. To assess split-half reliability of PFN functional topography, we conducted an additional analysis in which we leveraged data from ten participants who had at least twenty minutes of low-motion high-quality resting-state data. We first computed PFNs in each half of the data using the approach described in the main text and then calculated the split-half reliability for each participant and each PFN as the intraclass correlation coefficient (ICC) between the vertex-wise loadings in each half of the data. To contextualize our results: ICC scores greater than 0.9 are considered “excellent” while scores between 0.75 to 0.9 are considered “good”, scores between 0.5 and 0.75 are considered “moderate” and scores below 0.5 are considered “poor” (Koo and Li, 2017). Across all seventeen PFNs and all ten participants, all ICC scores fell within a range of 0.84 to 0.99, with the majority of ICC scores (96%) falling in the “excellent” category (ICC > 0.9). ICC scores for each participant and each network are shown in (a) a histogram of all ICC values, (b) scatterplots depicting ICC scores for each of the ten participants by PFN, and (c) a brain map depicting average ICC score by network.

Additionally, with regard to the reviewer’s last point about the quantity of data in this replication study as compared to the original study (Cui et al., 2020), we have now clarified that these are of comparable length in the manuscript on lines 537-541:

“We note that the original study¹⁶ utilized a concatenated time series from both resting-state and two task-based scans yielding a time series length of 27 minutes 45 seconds prior to preprocessing, while the current study uses up to four resting-state scans and three task-based scans yielding a maximum time series length of 29 minutes 36 seconds after motion scrubbing.”

4. Prediction accuracy of executive function and learning and memory. Given the NIH Toolbox data were broken into components using PCA, shouldn't we expect, by definition, that the first component would yield the highest prediction accuracy since it contains the greatest amount of between-person variance? To be clear, I am not saying this is not a valid or reproducible result; I am not sure what to make of the fact that these prediction accuracies are lower. It seems to just be epiphenomenal.

While it is true that the first principal component by definition explains the most between-person variance across cognitive tasks, this does not necessarily mean that this component will also have the strongest associations with another independent variable (in this case, PFN functional topography). This is because principal components analysis (PCA), unlike other techniques like partial least squares, does not fit using any information about the independent variable (Garthwaite, 1994). To confirm that our findings using the PCA approach (highest prediction accuracy for PC1: General Cognition, moderate prediction accuracy for PC3: Learning/Memory, and lowest prediction accuracy for PC2: Executive Function) are consistent with findings from an alternative approach, we re-ran our ridge regression models to predict performance on each cognitive task independently. This analysis revealed the highest prediction accuracies for the two tasks that loaded most highly on General Cognition, which was identified as the first component in the PCA. This was true for both the discovery (Oral Reading Test: $r = 0.34$, $p < .001$; Picture Vocabulary: $r = 0.38$, $p < .001$) and replication

(Oral Reading Test: $r = 0.39$, $p < .001$; Picture Vocabulary: $r = 0.40$, $p < .001$) samples. Similarly, we found the lowest prediction accuracies for the two tasks that had the greatest loading on the second principal component, Executive Function. This was true in both the discovery (Pattern Comparison: $r = 0.11$, $p < .001$; Flanker: $r = 0.13$, $p < .001$) and replication (Pattern Comparison: $r = 0.12$, $p < .001$; Flanker: $r = 0.13$, $p < .001$) samples. The two tasks that had the highest loading on the third principal component (Learning/Memory) fell in the middle for both the discovery (Picture Sequence: $r = 0.20$, $p < .001$; Rey Auditory and Verbal Learning: $r = 0.28$, $p < .001$) and replication (Picture Sequence: $r = 0.19$, $p < .001$; Rey Auditory and Verbal Learning: $r = 0.28$, $p < .001$) samples. This analysis, included in the new **Supplementary Table 5** copied below, confirms that the overall pattern of results we found using the PCA approach is consistent with results found using this alternative approach that does not make use of PCA.

Prediction Accuracy	Discovery		Replication	
	r	p	r	p
Picture Vocabulary	0.38	7.25×10^{-120}	0.40	4.55×10^{-132}
Oral Reading Test	0.34	3.39×10^{-93}	0.39	3.66×10^{-127}
List Sort	0.29	2.54×10^{-69}	0.32	5.87×10^{-82}
RAVLT	0.28	1.34×10^{-63}	0.28	4.18×10^{-64}
LMT	0.24	3.76×10^{-46}	0.24	1.93×10^{-47}
Card Sorting	0.22	9.24×10^{-39}	0.22	7.36×10^{-38}
Picture Sequence	0.20	1.85×10^{-31}	0.19	8.13×10^{-29}
Flanker Test	0.13	9.11×10^{-15}	0.13	5.54×10^{-14}
Pattern Comparison	0.11	3.06×10^{-10}	0.12	4.76×10^{-12}

Supplementary Table 5. Prediction accuracy of ridge regression models predicting performance on individual cognitive tasks. Multivariate prediction results from ridge regression models trained to predict cognitive performance from the multivariate pattern of PFN topography show comparable prediction accuracies (the correlation r between actual and predicted cognitive performance) as in our main results using PCA scores: the highest prediction accuracies are found for the two tasks that loaded most highly on the first principal component of General Cognition, while the lowest prediction accuracies were found for the two tasks loading highest on the second principal component of Executive Function and the two tasks loading highest on the third principal component of Learning/Memory fell in the middle across both the discovery and replication samples.

5. Similarly, the S-A axis results (Figure 5) also seem epiphenomenal. Since FPN shows a positive association with general cognitive ability and motor networks a negative relationship, does not a positive association along the S-A axis have to exist?

As a point of clarification, our S-A axis results (now **Figure 4**) describe the relationship between a given network's rank along the S-A axis and the prediction accuracy of a ridge regression model trained on the *multivariate* pattern of spatial topography of that network across all vertices. These data are independent of the *univariate* association results, in which the total cortical representation of FPN shows a positive association with general cognitive ability and motor networks show a negative relationship. While it is interesting to point out that the results of these two different approaches provide conceptual convergence, it is not necessarily the case that the multivariate prediction accuracies will align with the univariate association results. Indeed, a *strong negative* association such as we found for the motor networks might have implied a *higher* prediction accuracy than for networks with only weak association, since prediction accuracy does not depend on the direction of association.

Importantly, the goal of our analysis of S-A axis rank was to *describe* the spatial pattern of prediction accuracies across networks in order to understand how this heterogeneity aligns with a predominant axis of cortical variation (Sydnor et al., 2021). To confirm this spatial alignment was beyond what one might expect by chance, we used a conservative spin-based permutation test that demonstrated the true spatial alignment was significantly greater than the null distribution. In the revised manuscript, we have sought to clarify this point as well as to temper our explanation of the S-A axis analysis as being primarily descriptive of our prediction accuracy results on lines 260-263:

“These results demonstrate that a network's position along the S-A axis is associated with the relevance of its functional topography in predicting cognitive performance during childhood, providing a useful framework for describing and understanding the spatial pattern of prediction accuracy results across networks.”

REVIEWER #2

This manuscript investigates individual differences in cognition during childhood and attempts to demonstrate that personalized functional networks (PFNs) are associated with general cognition. The associations are then claimed to be validated using a ridge regression predictive model of cognition and the authors also show that prediction accuracy increases along the cortex's sensorimotor-association organizational atlas.

Strength of this work is that it replicates results published by this same group on slightly different data using slightly different measures, published in Neuron in 2020. Another strength is the use of personalized functional networks as input to both the association studies and as input to the predictive models.

Thank you for highlighting these strengths of our work.

1. Enthusiasm is reduced by the weak associations shown in figures 2 (b,c,d, f, g, h), and figure 4, yet much of the discussion and conclusions are focused on these associations.

While we recognize that the effect sizes in our univariate association results (now shown in revised Figure 5) are smaller than the results of our multivariate prediction analyses (now shown in Figures 2, 3 and 4), we do point out that our main multivariate prediction results have substantially *larger* effect sizes than have been found in prior work. Indeed, according to Marek, Tervo-Clemmens et al. (2022), effect sizes across three large datasets totaling over 35,000 scans show median correlation effect sizes between 0.02 to 0.03 with a range of less than -0.2 to 0.2. Our main findings (with effect sizes exceeding 0.4) are greater than would be expected in a typical analysis with a dataset of this size. Additionally, it is worth pointing out that the reported effect size of the multivariate analyses is nearly identical to our prior work (Cui et al., 2020), despite the fact that due to the “winner’s curse” replication analyses typically show smaller effect sizes than original studies (Patil et al., 2016).

In the revised manuscript, we address the reviewer’s concern that we have disproportionately emphasized the results of our univariate association results rather than our multivariate prediction results in three ways:

First, we have restructured our Results section to first focus on the predictive modeling results followed by the univariate association results.

Second, we have reduced the text in the discussion section that focuses on the univariate association results with small effect sizes, instead focusing more on our multivariate predictive modeling results on lines 327-332:

“In the largest study to use precision functional brain mapping to investigate cognition in children to date, we found reproducible associations between individual differences in functional brain network organization and individual differences in cognition. Replicating

key findings from a prior study¹⁶ in samples that were an order of magnitude larger, we trained cross-validated models on the complex multivariate pattern of personalized functional network topography to predict individual differences in cognitive functioning in unseen participants' data."

Third, we now contextualize the small effect sizes observed in the mass-univariate analysis by pointing out that they align with expected effect sizes for brain-behavior associations in datasets of this size, and remain highly reproducible across multiple large samples (lines 301-304):

"We also note that although the effect sizes for these univariate associations are small, they fall within or above the expected range for accurately-estimated brain-behavior effect sizes in studies of this size¹⁷ and these effects are highly reproducible across studies and samples."

2. The predictive models predict factors other than what is shown in the association studies and it's not clear why. For example this reviewer suspects it would be difficult to build a predictive model that works based on the associations shown in figure 2 (b,c,d) and (f,g,h) so then why show these associations? (e.g. total representation is not used in the predictive models)

We are happy to clarify. We conducted two separate analyses that we see as complementary: First, we conducted *univariate* association analyses to investigate the overall magnitude and directionality of associations between PFN topography and cognition using a simple summary metric (total cortical representation). This allowed us to replicate the finding from prior work (Cui et al., 2020) that the total cortical representation of fronto-parietal networks is positively associated with cognition, while the total cortical representation of motor networks is negatively associated with cognition. Second, we conducted *multivariate* prediction analyses, leveraging the full richness and complexity of vertex-wise functional topography for each PFN (see the schematic depicting this analysis in **Figure 1**). Our predictive models capitalized on this rich detail to make accurate predictions of cognitive performance in held-out data, again replicating prior results (Cui et al., 2020) in a much larger dataset. While the univariate association analyses tell us about how the overall representation of different networks are related to cognition, the multivariate analyses generate predictions of cognitive performance by taking into account each child's unique, complex pattern of functional network topography. We agree with the reviewer that it would be difficult to build a predictive model using total cortical representation, and such univariate prediction models are not ideal (Rosenberg and Finn, 2022); see also our response to comment #3 below.

3. The primary conclusions are focused on the associations with an acknowledgement that such association studies are not widely replicable (with reference to the Marek 2022 Nature paper) but no mention is made of the paper by Rosenberg and Finn (Nature Neuro 25(7), 2022) where they specifically point out that associations can be believed if one can build predictive models

from the data. Yet that is not what is done here. If the authors want to discuss associations between total representation and cognitive variables, then they should take those associations and build predictive models from them – yet this is not what is done.

The Marek et al argument of poor replicability can be found with weak associations even with large N, hence the need for predictive models on the direct associations – if they're not predictive, the associations are not meaningful ala the Rosenberg and Finn paper.

This warrants clarification. In our study, we conduct both *univariate association analyses* (**Figure 5**) to determine the directional relationship between total cortical representation of each PFN and cognitive performance, as well as *multivariate prediction analyses* (**Figures 2 and 3**) to predict cognitive performance in held-out data from complex patterns of PFN topography. As pointed out in the Rosenberg and Finn (2022) paper mentioned by the reviewer, an important recommendation for establishing robust brain-behavior relationships is the following:

“. . . we should focus on patterns of brain features rather than features in isolation. Marek et al. show improved reliability of multivariate patterns over univariate associations. This is perhaps unsurprising given that brain functions are inherently complex and interdependent, and our measures of them are inherently noisy; therefore, because ground-truth effect sizes for individual features are small, we get better signal when aggregating across many features. Another benefit to multivariate approaches is that they eschew the need to correct for multiple comparisons across features, which, as Marek et al. point out, can harm replication and generalizability by increasing false negatives...both theory and empirical evidence suggest that multivariate approaches are much more appropriate than mass univariate tests for establishing generalizable brain-behavior associations” (Rosenberg and Finn, 2022)

In line with this recommendation, our predictive modeling analyses make use of multivariate data (the pattern of functional topography for each PFN at each vertex) rather than univariate data (e.g., the summary metric of total cortical representation). Given that a primary goal of our study was to establish external validity by replicating and extend the results of prior work (Cui et al., 2020) in a new dataset (another important recommendation laid out by Rosenberg and Finn (2022)), we conducted both the univariate association analyses and multivariate prediction analyses in the same manner as the previous study (Cui et al., 2020). Our findings using both approaches replicated the findings of the prior study (Cui et al., 2020) in two matched halves of the data, directly addressing the reviewer's concern about the potential for replicability. To clarify this important point, we have included the following sentence in the revised methods section on lines 622-625:

“In line with the recommendation that predictive models of brain-behavior associations be trained on multivariate patterns rather than univariate measures (Rosenberg and Finn, 2022), these predictive models were trained on concatenated network loading matrices across the 17 PFNs.”

4. *The predictive models provided have outstanding performance, but it is unclear why, given the weak associations described. Many steps are taken between the associations shown however and the final predictive models that are built.*

To understand the outstanding performance of the predictive models described in **Figures 2 and 3** in light of the weak associations described in **Figure 5**, it is worth reiterating that the features differ substantially between these two separate analyses. First and foremost, the predictive models are trained on multivariate patterns of spatial topography across all vertices for each of the seventeen PFNs (see the schematic depicting this analysis in **Figure 1**). These models leverage the complex dimensionality from thousands of features per network. In contrast, the univariate associations are based on the total cortical representation (a summary metric) of each PFN individually – i.e., just one feature. By computing total cortical representation as the sum of weights for each PFN across all vertices for the association analyses, the rich detail regarding the functional topography that describes the spatial arrangement of these PFNs is lost. This detail is very likely to be important for the predictive models, which capitalize on this rich multivariate information to yield the best possible model performance. Furthermore, as noted above in response to comments 2 and 3 from this reviewer, it is not necessarily the case that multivariate prediction accuracy will align with these univariate association results. While it is interesting that the results of these two different approaches provide convergent results, there is no firm guarantee that the pattern of association strength will necessarily match the pattern of prediction accuracy. Indeed, a *strong negative* association such as we found for the motor networks might have implied a *higher* prediction accuracy than for networks with only weak association, since prediction accuracy does not depend on the direction of association. We have clarified this point in the revised manuscript on lines 608-614:

“We used linear mixed effects models (implemented with the “lme4” package in *R*) to assess associations between PFN topography (total cortical representation) and performance in each cognitive domain while accounting for both fixed and random predictors. We note that these univariate association analyses are distinct from our multivariate predictive modeling approach (using ridge regression, as described below) in that we use a single measure (total cortical representation) per network rather than the multivariate pattern of PFN topography across all vertices (i.e., thousands of features per network).”

a) How exactly was motion dealt with. Figure 2 shows results from a mixed effect analysis but it appears that motion played a significant role in this analysis. There should be more clarity on how age, sex, motion, and site were controlled for in this analysis.

The results depicted in original Figure 2 (now **Figure 5**) are from an analysis using linear mixed effects models to associate general cognition with the total cortical representation of our PFNs. Each model included covariates for age, sex, and head motion (quantified as mean framewise displacement), as well as a random effect for family ID (accounting for

siblings who participated in the ABCD study). As described in the revised manuscript on lines 293-294, 548-549 and 551-552 as well as in the **Figure 5** caption, data were harmonized across sites using ComBat (Fortin et al., 2017, 2018), thus obviating the need for a site covariate in this analysis. In addition to including motion as a covariate in all analyses, we also conservatively addressed this critical confound prior to this analysis in our rigorous inclusion criteria, excluding participants with fewer than 600 TRs remaining after motion censoring (frames exceeding a fractional displacement threshold of 0.2mm or failing to pass outlier detection at +/- 3 standard deviations). Finally, it should be noted that this data was preprocessed using a top-performing denoising pipeline that we have benchmarked in prior work (Ciric et al., 2017). This comprehensive strategy builds on our extensive prior work on motion artifact, denoising, and quality control over the past ten years (Ciric et al., 2018; Dosenbach et al., 2017; Satterthwaite, Elliott, et al., 2013; Satterthwaite et al., 2012, 2019; Satterthwaite, Wolf, et al., 2013).

We have clarified these important points on lines 608-618:

“We used linear mixed effects models (implemented with the “lme4” package in R) to assess associations between PFN topography (total cortical representation) and performance in each cognitive domain while accounting for both fixed and random predictors. We note that these univariate association analyses are distinct from our multivariate predictive modeling approach (using ridge regression, as described below) in that we use a single measure (total cortical representation) per network rather than the multivariate pattern of PFN topography across all vertices. All linear mixed effects models included fixed effects parameters for age, biological sex, head motion (mean fractional displacement), as well as a random intercept for family (accounting for siblings). Note that data were harmonized across sites prior to linear mixed-effects model analyses, and thus there was no need to include a random intercept for site.”

We have also provided clarification on lines 285-294:

“As previously,¹⁶ we first calculated the total cortical representation of each PFN as the sum of network loadings across all vertices, using the soft parcellation to account for spatial overlap across functional brain networks. As described in prior work,¹⁶ this measure of total cortical representation represents the spatial extent of each PFN on the cortical surface. We then applied linear mixed-effects models to probe the association between total cortical representation of each PFN and general cognition. These models accounted for age, sex (motivated by prior findings⁵⁶ that patterns of PFN total cortical representation differ by biological sex), family (to account for siblings in the ABCD dataset), and head motion (mean FD) as model covariates (**Table 1; Figure 5**). Multiple comparisons were accounted for using the Bonferroni method. Note that ComBat harmonization was applied to account for variability across sites.^{77,78}”

We also acknowledge the role of motion in our analyses as noted by the reviewer in the revised manuscript on lines 415-421:

“Second, head motion continues to pose an ongoing challenge for neuroimaging studies,^{58–60} particularly for studies of children,⁵⁸ and indeed our linear mixed effects models revealed that the total cortical representation of fronto-parietal PFNs was significantly associated with head motion. We have attempted to mitigate these effects in our analyses of interest by following best practices for reducing the influence of head motion on our results, including using a top-performing preprocessing pipeline and inclusion of motion as a covariate in all analyses.”

b) It appears to be the case that probabilistic PCA was performed on the cognitive measures and then predictive models built using the 1st, 2nd and 3rd PCs. If the PCA is performed on the entire sample before it was split into the discovery and replication set then the data is contaminated and this could account for the high prediction performance. Lines 471-474 seem to imply this was run on all the subjects together.

This is an important point that we are happy to address. We drew our cognitive measures from a PCA that was conducted in a prior study (W. K. Thompson et al., 2019), allowing our approach to be comparable with a large number of prior studies using these same site- and family-corrected principal components. However, the reviewer makes a valid point that since the Thompson et al. (2019) study computed the PCA on the full sample, this could potentially lead to leakage across our Discovery and Replication sub-samples. To address this point, we derived PCA components separately in the Discovery and Replication sub-samples and re-trained our predictive models to predict these independent (non-contaminated) cognitive scores. Repeating our main analyses with these new predictive models, we find nearly identical results as shown in the new **Supplementary Figure 5** copied below:

Supplementary Figure 5. Functional Topography Predicts Individual Differences in Cognitive Domains Derived by PCA in Independent Discovery and Replication Samples. To avoid contamination across discovery and replication samples in the cognitive outcome score that could lead to overfitting, we re-computed the cognitive domains of general cognition, executive function, and learning/memory using principal components analysis (PCA) conducted independently in the discovery and replication samples. Scatterplots depict the association between actual and predicted cognitive performance from ridge regression models trained to predict individual differences in general cognition (a), executive function (b), and learning/memory (c), using 2F-CV across both the discovery (black scatterplot) and replication (gray scatterplot) samples. Inset histograms represent the distributions of prediction accuracies from a permutation test. Brain maps depict the prediction accuracy results for ridge regression models trained to predict general cognition (d), executive function (e), or learning/memory (f) from the spatial topography of each PFN independently, with the highest prediction accuracies found in association cortex.

As one might expect, the correlation between our independent cognitive scores and the original cognitive scores from Thompson et al. (2019) are highly correlated (General Cognition/PC1: $r = 0.970$; Executive Function/PC2: $r = 0.985$; Learning and Memory/PC3: $r = 0.975$). This analysis provides strong evidence that our main results were not inflated by overfitting due to leakage. We have retained the results using the Thompson et al. (2019) cognitive scores in the main text of the manuscript to provide straightforward comparison with the substantial prior work using these scores, but have made note of this important limitation and our new analysis results on lines 649-654:

“To ensure that our results were not overfit as a result of leakage across samples by the cognitive outcome variables derived in the whole sample,⁴⁰ we also trained ridge

regression models to predict cognitive scores derived by two independent principal components analyses in the discovery and replication samples separately. Repeating our main analyses with these new predictive models, we find functionally identical results as shown in **Supplementary Figure 5.**”

c) Probably the best way to resolve this disconnect with the predictive models and the associations is to show the associations for the factors that were predicted or actually build the predictive model based on the associations under discussion.

We conducted *univariate* association analyses (using the summary metric of total cortical representation for each of the seventeen networks) and *multivariate* prediction analyses (using the high-dimensional pattern of functional topography for vertices in all seventeen networks; see **Figure 1**) rather than the other way around for several reasons: 1) These analyses provide complementary information: the association analyses provide information about the magnitude and directionality of associations between overall topography in each PFN and cognitive performance, while the predictive models demonstrate the accuracy with which cognitive performance can be predicted from the unique, complex pattern of functional network topography in each child; 2) These analyses were conducted as a replication and extension of prior work (Cui et al., 2020) and as such we undertook the same approach; and 3) To maximize generalizability of findings, we followed recommendations that predictive models be trained on multivariate patterns rather than univariate summary metrics (Rosenberg & Finn, 2022). This point is also explained in further detail above in response to comments #2 and #3.

5. Figure 3a and Figure 4a&e seem to fail the eye-ball test in that the line running through the data does not appear to follow the data. How are the lines derived and why don't they match the data? Was there a constraint placed on the regression to make it go through zero? (such a constraint would not really be valid here)

We did not impose any constraints on the regression to force them to go through zero. (the scatterplot points are centered on zero as a result of z-scoring these variables). Given the large number of participants in this study, it can be difficult to discern the density of points in these scatterplots. Therefore, to make it clearer that the simple correlation line plots do indeed follow the data, we have included the following new **Supplementary Figure 2** copied below depicting the results for each sample independently as hexplots.

Supplementary Figure 2. Hexplots of associations between actual and predicted cognitive performance from ridge regression models. Association between actual and predicted cognitive performance using two-fold cross-validation (2F-CV) with nested cross-validation for parameter tuning across both the discovery (top row) and replication (bottom row) samples, for predictions of General Cognition (first column), Executive Function (second column), and Learning/Memory (third column). Heatmap represents the density of points plotted in a given region.

This point is also clarified on lines 229-231:

“Associations between actual and predicted cognitive performance across all three tasks are depicted as hexplots in **Supplementary Figure 2** to further depict the density of points given the large number of participants in this study.”

REVIEWER #3

Keller & co-authors investigate how functional connectivity topography relate to cognitive and behavioral characteristics in a large sample of children (ABCD sample). Functional topography was characterized using a non-negative matrix factorization technique, to define 17 overlapping networks on each subject's cortical surface. Importantly, this technique allows for networks to be derived in a data-driven manner for each subject's functional data. The loadings for each of the 17 networks were summed and associated with a general cognitive trait, replicating a previously conducted analyses (although, with small effect sizes). Notably, this replication has been conducted in a much larger dataset than the previous study. Another replication was conducted using the network loadings, both combined and separated out, to predict cognition, executive function, and learning/memory. These analyses demonstrate the importance of data belonging to the fronto-parietal networks for these predictions. Finally, prediction accuracy was shown to rank correlate with the so-called S-A axis, placing these results within the larger context of studies of cortical gradients. The manuscript does well to highlight its contributions, namely that this investigation is a replication of previous work, with a larger sample and different preprocessing pipeline, and that the NMF approach allows for individualized representations of functional networks to be estimated cortex wide.

We thank the reviewer for highlighting these strengths of our study!

Main Critiques:

The results shown in Figure 2, use “total cortical representation” as the brain feature of interest. I find that more explanation, and possible exploration, of this feature should be provided for readers, to understand the initial result of this manuscript. Even in citation 16, it was not clear what this feature means. As I understand it, it is a sum of all NMF weights (for a specific network) across the whole cortex. I think readers might benefit from speculation of why greater the sum of these weights would relate positively to increased general cognition capacity. I am sympathetic to the idea that we don't know what it means at the biological level – and in fact, your study might provide evidence to suggest an interpretation of it: that greater representation is meaningful for prediction. To be clear, I think the authors provide excellent discussion of why the topography of fronto-parietal networks are important to predict these individual differences (the section, Association network topography supports domain-general cognitive abilities, is great). I'd like to see additional explanation that focuses on what “total cortical representation” is or what it conceptually represents (which could be additionally included in methods, even), since it is central to the manuscript.

We really appreciate this thoughtful remark. The reviewer is correct in their understanding of total cortical representation as a sum of all NMF weights for a specific network across the whole cortex. Following the reviewer's suggestion, we have added a clearer description of this measure to the Methods section on lines 596-600:

“For our univariate analyses, we calculated the total cortical representation of each PFN as the sum of network loadings across all vertices as previously,¹⁶ using the soft parcellation to account for spatial overlap across functional brain networks. As described in prior work,¹⁶ this measure of total cortical representation quantifies the spatial extent of each PFN on the cortical surface.”

To ensure that this point is clear throughout the manuscript, we have also added the following description of the interpretation of this measure on lines 287-288:

“As described in prior work,¹⁶ this measure of total cortical representation captures the spatial extent of each PFN on the cortical surface.”

To further describe the intuition for why the greater sum of these weights would relate positively to increased general cognition capacity, we now note in the revised results section on lines 295-297:

“We hypothesized that greater total cortical representation of association networks would be associated with better general cognitive abilities, in line with the intuition that more cortical surface space dedicated to these networks would facilitate the functions they subserve.”

Relatedly, I could imagine that there might be some sources of potential bias, related to head size for the total cortical representation feature. Individual's surfaces need to be stretched and squeezed to the common fsLR surface. What I am getting at is the possibility that cortical representation of a specific brain network could be influenced by this transformation from individual to common surface space. There is the possibility that a hypothetical functional area that initially occupied little cortical surface in subject space, has the potential to be expanded, if that area needs to be stretched, and thus, the functional data interpolated to occupy a larger space to match fsLR. I am not sure the NMF clustering on this data would mitigate this necessarily. Have you investigated ways to modulate the loading variables by some stretching/squeezing feature? Akin to how in Voxel-Based Morphometry, grey matter probability values are scaled by Jacobian determinants to account for local compression and stretching after the grey matter probability values that have been warped to the common space. If this computational undertaking is unfeasible for the authors, (and the authors agree with the logic of my critique) I imagine this could be noted as a limitation. This critique only affects mainly “total cortical representation” feature. Although, I do wonder how Prediction Accuracy is related to the original “size” (but of course, these networks actually have loadings across the cortex) of each network map (referring to the visualization in Figure 1e); just eyeballing the results, it looks like the least predictive networks (4, 13, 15, and 6) cover much less space than the most predictive ones (3, 9, and 12). It might be useful to plot the prediction accuracy versus network “size” (however you might define that—you might need to pick a threshold on the weights) and include that as a supplementary figure.

We agree that this is a potential limitation that is outside the scope of this paper due to computational feasibility (lines 430-437):

“Fifth, the registration of MRI data to a common reference space (*fslr*) has known limitations (e.g., the spatial warping to register individual brains with this common reference will necessarily differ across individuals). While this approach had the advantage of allowing us to apply non-negative matrix factorization with a common spatial prior across individuals (allowing us to account for individual variability in functional neuroanatomy in a standardized space), future work may investigate whether the varying degrees of stretching/squeezing of different cortical surface regions has an effect on associations between functional topography and behavior.”

Additionally, to address the reviewer’s point about the relationship between PFN size and prediction accuracy, we have included a new **Supplementary Figure 6** copied below depicting the relationship between the size of each network and prediction accuracy. We defined size as the sum of vertices designated as belonging to each PFN in our hard parcellation (designations were made based on the network with the highest loading at each vertex).

Supplementary Figure 6. Prediction accuracy and S-A axis rank by PFN size.

Prediction accuracies by network (the correlation r between actual and predicted cognitive performance) from ridge regression models trained on the vertex-wise pattern of topography for each PFN to predict (a) General Cognition, (b) Executive Function, or (c) Learning/Memory are plotted on the y-axes. PFN sizes, defined as the number of vertices belonging to each PFN in the hard parcellation, are plotted on the x-axes. The correlations between PFN size and prediction accuracy are significant for all three cognitive domains (General Cognition: $r = 0.80$, $p = 0.0001$; Executive Function: $r = 0.52$, $p = 0.032$; Learning/Memory: $r = 0.57$, $p = 0.017$). (d) Association between sensorimotor-association (S-A) axis rank and PFN size ($r = 0.44$, $p = 0.077$).

As the reviewer suggested, there was indeed a significant relationship between the size of each PFN and its prediction accuracy for all three domains of cognition assessed. We also found that PFN size tends to increase along the S-A axis in accordance with prior results (Power et al., 2013), although the correlation between PFN size and the average S-A axis rank across all vertices within each PFN was non-significant ($r = 0.44$, $p = 0.077$). This trend suggests that in general, association networks that predict cognition also tend to

span more cortical surface area than sensorimotor networks. We now include a discussion of this point in the main text on lines 656-662:

“We also tested whether the prediction accuracy from these ridge regression models was associated with the total size of each PFN, defined as the number of vertices belonging to each PFN in the hard parcellation (**Supplementary Figure 6**). However, we note that association networks ranked higher along the S-A axis tend to occupy a larger portion of the cortical surface than sensorimotor networks ranked lower along the axis (see following section on Associations Between Prediction Accuracy and S-A Axis Rank).”

I understand (and appreciate) that the authors used state-of-the-art processing pipelines, have already eliminated thousands of subjects based on motion, and have censored the time series. However, given that meanFD is found to be a significant term in the mixed effects model for each of the PFN networks, in both the replication and discovery cohorts (Table 1 results), I'll provide one more suggestion. Since data is combined from multiple runs (rest and three task scans), is there an effect of motion that could be localized to each run. In other words, could # censored frames, for example, for each run be included as 4 separate nuisance covariates in the model. This might account for differences in participant vigilance (or restlessness) in the context of rest versus the different tasks, that could interact with the associations to the cognitive and behavioral variables.

We appreciate the reviewer's recognition that we have used state-of-the-art top-performing processing pipelines, have applied strict exclusion criteria for head motion, and have included motion as a covariate in all of our analyses. We take the reviewer's point that head motion (meanFD) was significantly associated with the total cortical representation of all three fronto-parietal PFNs in our linear mixed effects models. This is an important point, which we have now added to the discussion section on lines 415-421 to aid interpretation:

“Second, head motion continues to pose an ongoing challenge for neuroimaging studies,⁵⁸⁻⁶⁰ particularly for studies of children,⁵⁸ and indeed our linear mixed effects models revealed that the total cortical representation of fronto-parietal PFNs was significantly associated with head motion. We have attempted to mitigate these effects in our analyses of interest by following best practices for reducing the influence of head motion on our results, including using a top-performing preprocessing pipeline and inclusion of motion as a covariate in all analyses.”

Since each of the thousands of individual participants has a variable number of frames remaining per run following our strict motion correction, it is not feasible to control for head motion independently for each run. Moreover, we feel it is more accurate to correct for motion across the full concatenated time series, since this is the form of the data that we use for all of our analyses. By correcting for motion across the concatenated time series across all runs and using the overall mean fractional displacement as a covariate in our analyses, we avoid inadvertently over- or under-correcting for head motion for

certain runs due to their variable lengths. We now clarify this point in the revised manuscript on lines 542-547:

“Since each of the participants has a variable number of frames remaining per run following our strict motion correction, it is not feasible to control for head motion independently for each run. Instead, controlling for mean fractional displacement across the concatenated time series provides a more accurate motion correction and better reflects the data we use in all of our analyses. We therefore use only the concatenated time series across all runs and the overall mean fractional displacement as a motion covariate in our analyses, so as not to inadvertently over- or under-correct for head motion for certain runs.”

Minor comments:

For Figure 2, it would be nice to make the axes have the same range in (a) and (e), to make comparisons easier on the eye.

We agree and have revised Figure 2 (now **Figure 5**) to have the axes in the same range in (a) and (e) as suggested.

In the limitations it was noted that data from two (but I thought it was three, as noted in the methods...) were task regressed, but the methods do not detail how the task regression was performed.

We have corrected this typo in the text, which was mistakenly drawn from prior work (Cui et al., 2020). We indeed used three behavioral tasks (not two) in our analyses and have removed the phrase about task regression to properly reflect the methods used in our study, which did not include task regression. Furthermore, in line with prior work (Hermosillo et al., 2022; Gratton et al., 2018) showing that the specific tasks performed during scanning do not strongly influence functional connectivity networks, we have confirmed that all of our key results remain significant when PFNs are derived using resting-state data only ($n=5,968$ participants included who had sufficient TRs remaining in the rest-only data) as a sensitivity analysis. First, linear mixed effects models confirm that the total cortical representation of fronto-parietal networks 3, 15 and 17 are positively associated with general cognition across the discovery (PFN 3: $\beta = 0.17$, $p_{\text{bonf}} = 2.16 \times 10^{-9}$; PFN 15: $\beta = 0.05$, $p_{\text{bonf}} = 0.074$; PFN 17: $\beta = 0.08$, $p_{\text{bonf}} = 0.003$) and replication (PFN 3: $\beta = 0.21$, $p_{\text{bonf}} = 1.60 \times 10^{-15}$; PFN 15: $\beta = 0.10$, $p_{\text{bonf}} = 1.74 \times 10^{-4}$; PFN 17: $\beta = 0.08$, $p_{\text{bonf}} = 0.004$) samples. Second, our ridge regression models used to predict cognitive performance from the multivariate pattern of PFN topography show

comparable prediction accuracy as in our main results for both the discovery (General Cognition: $r = 0.41$, $p < .001$; Executive Function: $r = 0.16$, $p < .001$; Learning/Memory: $r = 0.26$, $p < .001$) and replication (General Cognition: $r = 0.41$, $p < .001$; Executive Function: $r = 0.12$, $p < .001$; Learning/Memory: $r = 0.25$, $p < .001$) samples. We describe this approach in the revised manuscript on lines 537-537:

“Following preprocessing, we concatenated the time series data for both resting-state scans and three task-based scans (Monetary Incentive Delay Task, Stop-Signal Task, and Emotional N-Back Task) as in prior work¹⁶ to maximize the available data for our analyses, though we note that all of our main results hold when we derive PFNs using only resting-state data from a subset of participants ($n=5,968$) who met our minimum threshold for clean resting-state data (**Supplementary Table 4**).”

a. Univariate Association Results

Predictors	PFN 3				PFN 15				PFN 17			
	β	Std. Error	t	p_{bonf}	β	Std. Error	t	p_{bonf}	β	Std. Error	t	p_{bonf}
Discovery												
Intercept	0.02	0.03	0.71	0.476	0.07	0.03	2.48	0.013	-0.03	0.03	-1.06	0.289
Age	0.01	0.02	0.28	0.778	-0.02	0.02	-0.86	0.388	-0.04	0.02	-1.84	0.065
Sex	-0.07	0.04	-1.77	0.076	-0.16	0.04	-4.10	4.26 x 10⁻⁵	0.05	0.04	1.30	0.195
Mean FD	-0.13	0.02	-6.71	2.42 x 10⁻¹¹	0.17	0.02	8.67	7.13 x 10⁻¹⁸	-0.11	0.02	-5.57	2.84 x 10⁻⁸
General Cognition	0.17	0.03	6.01	2.16 x 10⁻⁹	0.05	0.03	1.79	0.074	0.08	0.03	2.97	0.003
Replication												
Intercept	0.04	0.03	1.33	0.183	0.05	0.03	1.96	0.051	0.01	0.03	0.32	0.747
Age	-0.01	0.02	-0.45	0.649	-0.02	0.02	-0.99	0.323	-0.01	0.02	-0.46	0.646
Sex	-0.08	0.04	-2.24	0.025	-0.13	0.04	-3.37	7.65 x 10⁻⁴	-0.01	0.04	-0.15	0.881
Mean FD	-0.14	0.02	-7.78	1.01 x 10⁻¹⁴	0.14	0.02	7.67	2.36 x 10⁻¹⁴	-0.05	0.02	-2.76	0.006
General Cognition	0.21	0.03	8.01	1.60 x 10⁻¹⁵	0.10	0.03	3.76	1.74 x 10⁻⁴	0.08	0.03	2.85	0.004

b. Multivariate Prediction Results

Prediction Accuracy	Discovery		Replication	
	r	p	r	p
General Cognition	0.41	7.96 x 10 ⁻¹⁰⁸	0.41	5.62 x 10 ⁻¹¹⁵
Executive Function	0.16	2.63 x 10 ⁻¹⁶	0.12	3.02 x 10 ⁻¹¹
Learning/Memory	0.26	7.38 x 10 ⁻⁴³	0.25	2.47 x 10 ⁻⁴⁰

Supplementary Table 4. Replication of our main analyses using PFNs derived from resting-state data only ($n=5,968$ participants included who had sufficient TRs remaining in the rest-only data). **(a)** Univariate association analyses using linear mixed effects models confirm that the total cortical representation of fronto-parietal PFNs 3, 15 and 17 are positively associated with general cognition across the discovery and replication samples. **(b)** Multivariate prediction results from ridge regression models trained to predict cognitive performance from the multivariate pattern of PFN topography show comparable prediction accuracy (the correlation r between actual and

predicted cognitive performance) as in our main results for both the discovery and replication samples.

REVIEWER #4

Thank you for the opportunity to review the manuscript Personalized Functional Brain Network Topography Predicts Individual Differences in Youth Cognition. The study is well-written, interesting, and timely. Statistical methodology is robust and leverages a large-scale dataset from the ABCD study. The study examines inter-individual heterogeneity in functional brain network organization using functional mapping and cross-validated models. The authors report three frontoparietal PFNs that were significantly associated with general cognition across both discovery and replication samples, with a somatomotor network inversely associated with cognitive performance. The study explores a much needed area of research in functional connectivity, inter-individual heterogeneity in functional network topography, and brain-wide association studies. However, it is largely a replication of Cui et al., (2020) with the ABCD dataset. Nonetheless, the study is elegant and important to the field.

We thank the reviewer for highlighting these strengths of our study!

1. The authors concatenated across resting-state and task fMRI. This approach is understandable to provide a sufficient voxel timecourse for connectivity analyses. However, I have two concerns. First, nuanced detail is lost regarding potential differences in resting-state vs task-based fMRI in predicting general cognition or subdomains of cognition. Second, the three tasks are related but distinct and capture different aspects of cognition and emotion processing. There is an opportunity here to provide more nuanced detail of PFNs, that builds on but also extends the initial study from Cui et al. (2020). It would be helpful if the authors conducted additional follow up, sub-analyses for resting-state vs task to examine associations between PFNs and cognition. In addition, to fully test replication with Cui et al., which leveraged resting-state and tasks of emotion perception and n-back, I would like to see a sub-analysis repeated with resting-state and conceptually similar tasks such as n-back from ABCD.

As suggested by the reviewer, we have addressed this point in our revised manuscript by replicating our analyses in PFNs derived from resting-state data only ($n=5,968$ participants included who had sufficient TRs remaining in the rest-only data) as a sensitivity analysis. Our results replicate well using this subset of the data. First, linear mixed effects models confirm that the total cortical representation of fronto-parietal networks 3, 15 and 17 are positively associated with general cognition across the discovery (PFN 3: $\beta = 0.17$, $p_{\text{bonf}} = 2.16 \times 10^{-9}$; PFN 15: $\beta = 0.05$, $p_{\text{bonf}} = 0.074$; PFN 17: $\beta = 0.08$, $p_{\text{bonf}} = 0.003$) and replication (PFN 3: $\beta = 0.21$, $p_{\text{bonf}} = 1.60 \times 10^{-15}$; PFN 15: $\beta = 0.10$, $p_{\text{bonf}} = 1.74 \times 10^{-4}$; PFN 17: $\beta = 0.08$, $p_{\text{bonf}} = 0.004$) samples. Second, our ridge regression models used to predict cognitive performance from the multivariate pattern of PFN topography show comparable prediction accuracy as in our main results for both the discovery (General Cognition: $r = 0.41$, $p < .001$; Executive Function: $r = 0.16$, $p < .001$; Learning/Memory: $r = 0.26$, $p < .001$) and replication (General Cognition: $r = 0.41$,

$p < .001$; Executive Function: $r = 0.12$, $p < .001$; Learning/Memory: $r = 0.25$, $p < .001$) samples. We describe this approach in the revised manuscript on lines 532-537:

“Following preprocessing, we concatenated the time series data for both resting-state scans and three task-based scans (Monetary Incentive Delay Task, Stop-Signal Task, and Emotional N-Back Task) as in prior work¹⁶ to maximize the available data for our analyses, though we note that all of our main results hold when we derive PFNs using only resting-state data from a subset of participants ($n=5,968$) who met our minimum threshold for clean resting-state data (**Supplementary Table 4**).”

a. Univariate Association Results

Predictors	PFN 3				PFN 15				PFN 17			
	β	Std. Error	t	p_{bonf}	β	Std. Error	t	p_{bonf}	β	Std. Error	t	p_{bonf}
Discovery												
Intercept	0.02	0.03	0.71	0.476	0.07	0.03	2.48	0.013	-0.03	0.03	-1.06	0.289
Age	0.01	0.02	0.28	0.778	-0.02	0.02	-0.86	0.388	-0.04	0.02	-1.84	0.065
Sex	-0.07	0.04	-1.77	0.076	-0.16	0.04	-4.10	4.26 x 10⁻⁵	0.05	0.04	1.30	0.195
Mean FD	-0.13	0.02	-6.71	2.42 x 10⁻¹¹	0.17	0.02	8.67	7.13 x 10⁻¹⁸	-0.11	0.02	-5.57	2.84 x 10⁻⁸
General Cognition	0.17	0.03	6.01	2.16 x 10⁻⁹	0.05	0.03	1.79	0.074	0.08	0.03	2.97	0.003
Replication												
Intercept	0.04	0.03	1.33	0.183	0.05	0.03	1.96	0.051	0.01	0.03	0.32	0.747
Age	-0.01	0.02	-0.45	0.649	-0.02	0.02	-0.99	0.323	-0.01	0.02	-0.46	0.646
Sex	-0.08	0.04	-2.24	0.025	-0.13	0.04	-3.37	7.65 x 10⁻⁴	-0.01	0.04	-0.15	0.881
Mean FD	-0.14	0.02	-7.78	1.01 x 10⁻¹⁴	0.14	0.02	7.67	2.36 x 10⁻¹⁴	-0.05	0.02	-2.76	0.006
General Cognition	0.21	0.03	8.01	1.60 x 10⁻¹⁵	0.10	0.03	3.76	1.74 x 10⁻⁴	0.08	0.03	2.85	0.004

b. Multivariate Prediction Results

Prediction Accuracy	Discovery		Replication	
	r	p	r	p
General Cognition	0.41	7.96 x 10 ⁻¹⁰⁸	0.41	5.62 x 10 ⁻¹¹⁵
Executive Function	0.16	2.63 x 10 ⁻¹⁶	0.12	3.02 x 10 ⁻¹¹
Learning/Memory	0.26	7.38 x 10 ⁻⁴³	0.25	2.47 x 10 ⁻⁴⁰

Supplementary Table 4. Replication of our main analyses using PFNs derived from resting-state data only ($n=5,968$ participants included who had sufficient TRs remaining in the rest-only data). **(a)** Univariate association analyses using linear mixed effects models confirm that the total cortical representation of fronto-parietal PFNs 3, 15 and 17 are positively associated with general cognition across the discovery and replication samples. **(b)** Multivariate prediction results from ridge regression models trained to predict cognitive performance from the multivariate pattern of PFN topography show comparable prediction accuracy (the correlation r between actual and predicted cognitive performance) as in our main results for both the discovery and replication samples.

Regarding the reviewer's point about the alignment between the tasks used in the ABCD study (Emotional N-Back, Monetary Incentive Delay, and Stop-Signal Task) with the tasks used in the original study by Cui et al. (2020) (N-Back and Emotion Identification), we were not able to control for the fact that different tasks were used in different studies. However, we note that this replication of our results (with very similar effect size) across these studies highlights what has been previously shown in prior work (Gratton et al., 2018, *Neuron*) -- that the specific tasks performed during scanning do not dominate functional connectivity networks. Finally, it should be noted that because of the computational demands of the NMF algorithm and the extreme size of the ABCD sample, the replication of the results involving NMF entails exceptionally large computational requirements. While we were able to accomplish the above analysis on resting-state data only by capitalizing on one of the largest imaging-dedicated academic compute clusters in the world (e.g., CUBIC: <https://www.med.upenn.edu/cbica/cubic.html>), it unfortunately is not feasible to repeat this analysis on multiple separate subsets of the data (i.e., each individual task) as suggested by the reviewer. We hope that the existing literature and the new convergent results presented here from the rest-only timeseries adequately addresses the reviewer's remark.

2. One strength of the ABCD study for brain-wide association studies and fMRI is the sample heterogeneity. However, because the dataset includes youth with a range of psychiatric symptom severities, the number of children taking psychotropic medications in the discovery/replication samples should also be reported and accounted for in models. This is essential given the effect of psychotropic medications on functional connectivity and the heterogeneity of the ABCD sample.

We agree with the reviewer that an important strength of the ABCD study for brain-wide association studies and fMRI is the rich sample heterogeneity. To address the point that these participants may be taking psychotropic medications, we have included this information in the revised demographics table copied below. Importantly, we show that there are no significant differences between the discovery and replication samples in the use of antidepressant, antipsychotic, or ADHD medications (assessed by the Medication Inventory from the PhenX instrument and coded as in (Shoval et al., 2021)).

	Total	Discovery	Replication	P-value
Age (Months)	119.5 (±7.5)	119.5 (±7.6)	119.5 (±7.5)	0.94
Sex (F)	3,494 (50.1%)	1,806 (51.2%)	1,688 (49.0%)	0.062
General Cognition	0.1 (±0.7)	0.1 (±0.7)	0.1 (±0.7)	0.18
Executive Function	0.0 (±0.8)	0.0 (±0.8)	0.1 (±0.7)	0.23
Learning/Memory	0.1 (±0.7)	0.1 (±0.7)	0.0 (±0.7)	0.22
ADHD Medications	536 (7.7%)	283 (8.0%)	253 (7.3%)	0.30
Antidepressant Medication	122 (1.8%)	69 (2.0%)	53 (1.5%)	0.20
Antipsychotic Medication	40 (0.6%)	23 (0.7%)	17 (0.5%)	0.43
CBCL Ext.				0.93
Mean (SD)	4.2 (±5.5)	4.2 (±5.6)	4.1 (±5.4)	
Missing	1 (0.0%)	0 (0%)	1 (0.0%)	
CBCL Int.				0.50
Mean (SD)	5.0 (±5.5)	5.0 (±5.6)	4.9 (±5.4)	
Missing	1 (0.0%)	0 (0%)	1 (0.0%)	
CBCL Prob.				0.99
Mean (SD)	17.3 (±17.2)	17.4 (±17.5)	17.1 (±16.9)	
Missing	1 (0.0%)	0 (0%)	1 (0.0%)	
Household Income				0.28
[<50K]	1,645 (23.6%)	806 (22.9%)	839 (24.4%)	
[>=50K & <100K]	1,902 (27.3%)	983 (27.9%)	919 (26.7%)	
[>=100K]	2,870 (41.2%)	1,451 (41.2%)	1,419 (41.2%)	
Missing	552 (7.9%)	284 (8.1%)	268 (7.8%)	
Race				0.46
White	4,723 (67.8%)	2,426 (68.8%)	2,297 (66.7%)	
Black	910 (13.1%)	442 (12.5%)	468 (13.6%)	
Asian	145 (2.1%)	71 (2.0%)	74 (2.1%)	
AIAN/NHPI	41 (0.6%)	23 (0.7%)	18 (0.5%)	
Other	242 (3.5%)	114 (3.2%)	128 (3.7%)	
Mixed	823 (11.8%)	410 (11.6%)	413 (12.0%)	
Missing	85 (1.2%)	38 (1.1%)	47 (1.4%)	
Parent Education				0.31
< HS Diploma	236 (3.4%)	117 (3.3%)	119 (3.5%)	
HS Diploma/GED	542 (7.8%)	254 (7.2%)	288 (8.4%)	
Some College	1,781 (25.6%)	905 (25.7%)	876 (25.4%)	
Bachelor	1,885 (27.0%)	980 (27.8%)	905 (26.3%)	
Post Graduate Degree	2,519 (36.1%)	1,264 (35.9%)	1,255 (36.4%)	
Missing	6 (0.1%)	4 (0.1%)	2 (0.1%)	

Table 2. Demographic characteristics and variables of interest in the matched discovery ($n=3,525$) and replication ($n=3,447$) samples. Acronyms: AIAN = American Indian/Alaska Native; NHPI = Native Hawaiian and other Pacific Islander; HS = High School; GED = General Educational Development; CBCL = Child Behavior Checklist.

We also find that our results remain significant when controlling for psychotropic medication use as a covariate in our linear mixed effects models associating total cortical representation of fronto-parietal PFNs with general cognition in both the discovery (PFN 3: $\beta = 0.08$, $p_{\text{bonf}} = 6.69 \times 10^{-4}$; PFN 15: $\beta = 0.09$, $p_{\text{bonf}} = 4.87 \times 10^{-4}$; PFN 17: $\beta = 0.11$, $p_{\text{bonf}} = 1.07 \times 10^{-5}$) and replication (PFN 3: $\beta = 0.07$, $p_{\text{bonf}} = 0.003$; PFN 15: $\beta = 0.09$, $p_{\text{bonf}} = 2.63 \times 10^{-4}$; PFN 17: $\beta = 0.12$, $p_{\text{bonf}} = 2.66 \times 10^{-6}$) samples. The new **Supplementary Table 2** is copied below.

Predictors	PFN 3				PFN 15				PFN 17			
	β	Std. Error	t	p_{bonf}	β	Std. Error	t	p_{bonf}	β	Std. Error	t	p_{bonf}
Discovery												
Intercept	0.49	0.22	2.20	0.028	-0.10	0.22	-0.44	0.659	0.18	0.23	0.79	0.428
Age	-0.05	0.02	-2.62	0.009	-0.02	0.02	-1.27	0.206	-0.05	0.02	-2.62	0.009
Sex	-0.06	0.03	-1.78	0.075	-0.15	0.03	-4.53	6.17×10^{-6}	0.05	0.03	1.36	0.174
Mean FD	0.12	0.02	7.02	2.56×10^{-12}	0.11	0.02	6.78	1.38×10^{-11}	0.04	0.02	2.25	0.025
ADHD Meds	-0.04	0.07	-0.53	0.594	-0.05	0.07	-0.81	0.419	-0.08	0.07	-1.20	0.231
Antipsychotic Meds	-0.36	0.22	-1.65	0.099	0.22	0.21	1.03	0.304	-0.24	0.22	-1.11	0.267
Antidepressant Meds	-0.09	0.13	-0.71	0.475	-0.00	0.13	-0.01	0.991	0.10	0.13	0.80	0.421
General Cognition	0.08	0.02	3.41	6.69×10^{-4}	0.09	0.02	3.49	4.87×10^{-4}	0.11	0.02	4.41	1.07×10^{-5}
Replication												
Intercept	0.04	0.26	0.16	0.871	-0.18	0.26	-0.68	0.498	0.12	0.26	0.47	0.637
Age	-0.01	0.02	-0.81	0.418	-0.06	0.02	-3.29	0.001	-0.05	0.02	-2.68	0.007
Sex	-0.04	0.03	-1.16	0.245	-0.15	0.03	-4.18	2.94×10^{-5}	0.04	0.03	1.06	0.287
Mean FD	0.15	0.02	8.98	4.27×10^{-19}	0.08	0.02	4.53	6.19×10^{-6}	0.04	0.02	2.11	0.035
ADHD Meds	0.12	0.07	1.82	0.069	0.02	0.07	0.27	0.791	-0.01	0.07	-0.13	0.896
Antipsychotic Meds	-0.15	0.25	-0.62	0.534	0.34	0.24	1.40	0.163	0.06	0.25	0.23	0.817
Antidepressant Meds	0.00	0.14	0.03	0.977	-0.11	0.14	-0.80	0.423	-0.20	0.14	-1.41	0.159
General Cognition	0.07	0.02	2.97	0.003	0.09	0.03	3.65	2.63×10^{-4}	0.12	0.02	4.70	2.66×10^{-6}

Supplementary Table 2. Sensitivity analyses controlling for psychotropic medication use.

Linear mixed effects models associating general cognition with the total cortical representation of fronto-parietal PFNs remain significant across both the discovery and replication samples when controlling for psychotropic medication use (assessed by the Medication Inventory from the PhenX instrument and coded as in Shoval et al., 2021; though we note that it is not clear from these measures whether psychotropic medications were taken on the same day as the neuroimaging assessments).

This finding is now noted in the main text on lines 297-301:

“We found that all three fronto-parietal PFNs (networks 3, 15, and 17) were significantly positively associated with general cognition across both the discovery and replication samples. These findings remained significant in sensitivity analyses controlling for psychotropic medication use (**Supplementary Table 2**)”

We chose to present this analysis in a supplementary table rather than in the main text due to known problems with the assessment of psychotropic medication use in this sample (critically, it is not clear whether psychotropic medications were taken on the same day as the neuroimaging assessments, and thus the impact of drug use on functional imaging metrics cannot be specifically addressed). This point is now noted in the revised supplementary material.

3. Related to above, were there differences between the discovery/replication samples in terms of child psychopathology? The authors may want to consider testing for differences between the samples using the CBCL externalizing and internalizing scales, or the CBCL overall problem behavior scale. The authors conducted random cross-validation, which is great. However, additional follow-up analyses are essential for interpretation to demonstrate findings are not impacted by severity of psychopathology among discovery and replication samples.

There were no significant differences between the discovery and replication samples in terms of child psychopathology. In line with the reviewer's suggestion, we tested for differences between the samples on the CBCL externalizing, internalizing, and problem behavior scales and found no significant differences. These results are included in the revised demographics table copied below. Furthermore, as noted by the reviewer, the highly convergent results from repeated random split-half cross-validation demonstrates that differences in psychopathology across discovery and replication samples (or training and testing samples) did not impact our results. Moreover, an independent study from our laboratory focused specifically on relating psychopathology to PFN topography in youth has already been conducted (see Cui et al., 2022),

	Total	Discovery	Replication	P-value
Age (Months)	119.5 (±7.5)	119.5 (±7.6)	119.5 (±7.5)	0.94
Sex (F)	3,494 (50.1%)	1,806 (51.2%)	1,688 (49.0%)	0.062
General Cognition	0.1 (±0.7)	0.1 (±0.7)	0.1 (±0.7)	0.18
Executive Function	0.0 (±0.8)	0.0 (±0.8)	0.1 (±0.7)	0.23
Learning/Memory	0.1 (±0.7)	0.1 (±0.7)	0.0 (±0.7)	0.22
ADHD Medications	536 (7.7%)	283 (8.0%)	253 (7.3%)	0.30
Antidepressant Medication	122 (1.8%)	69 (2.0%)	53 (1.5%)	0.20
Antipsychotic Medication	40 (0.6%)	23 (0.7%)	17 (0.5%)	0.43
CBCL Ext.				0.93
Mean (SD)	4.2 (±5.5)	4.2 (±5.6)	4.1 (±5.4)	
Missing	1 (0.0%)	0 (0%)	1 (0.0%)	
CBCL Int.				0.50
Mean (SD)	5.0 (±5.5)	5.0 (±5.6)	4.9 (±5.4)	
Missing	1 (0.0%)	0 (0%)	1 (0.0%)	
CBCL Prob.				0.99
Mean (SD)	17.3 (±17.2)	17.4 (±17.5)	17.1 (±16.9)	
Missing	1 (0.0%)	0 (0%)	1 (0.0%)	
Household Income				0.28
[<50K]	1,645 (23.6%)	806 (22.9%)	839 (24.4%)	
[≥50K & <100K]	1,902 (27.3%)	983 (27.9%)	919 (26.7%)	
[≥100K]	2,870 (41.2%)	1,451 (41.2%)	1,419 (41.2%)	
Missing	552 (7.9%)	284 (8.1%)	268 (7.8%)	
Race				0.46
White	4,723 (67.8%)	2,426 (68.8%)	2,297 (66.7%)	
Black	910 (13.1%)	442 (12.5%)	468 (13.6%)	
Asian	145 (2.1%)	71 (2.0%)	74 (2.1%)	
AIAN/NHPI	41 (0.6%)	23 (0.7%)	18 (0.5%)	
Other	242 (3.5%)	114 (3.2%)	128 (3.7%)	
Mixed	823 (11.8%)	410 (11.6%)	413 (12.0%)	
Missing	85 (1.2%)	38 (1.1%)	47 (1.4%)	
Parent Education				0.31
< HS Diploma	236 (3.4%)	117 (3.3%)	119 (3.5%)	
HS Diploma/GED	542 (7.8%)	254 (7.2%)	288 (8.4%)	
Some College	1,781 (25.6%)	905 (25.7%)	876 (25.4%)	
Bachelor	1,885 (27.0%)	980 (27.8%)	905 (26.3%)	
Post Graduate Degree	2,519 (36.1%)	1,264 (35.9%)	1,255 (36.4%)	
Missing	6 (0.1%)	4 (0.1%)	2 (0.1%)	

Table 2. Demographic characteristics and variables of interest in the matched discovery ($n=3,525$) and replication ($n=3,447$) samples. Acronyms: AIAN = American Indian/Alaska Native; NHPI = Native Hawaiian and other Pacific Islander; HS = High School; GED = General Educational Development; CBCL = Child Behavior Checklist.

4. The authors may also want to consider greater consideration for race/ethnicity in sub-analyses. An advantage of large-scale imaging datasets, such as the ABCD study, is the opportunity to explore differential effects of functional connectivity that may be influenced by race/ethnicity. Similarly, it would be helpful to test if there is an association between network topography and sex.

While we agree with the reviewer that analyses of race/ethnicity are critically important, we feel that such an undertaking is outside the scope of the present study. In line with the editor's comment pasted below, we feel that there is not sufficiently strong justification for conducting these analyses in our current manuscript:

“If you decide to include these analyses please--in line with our policies--ensure that you provide strong justifications for doing so. If you do not have a strong justification (if there is no reason to expect a relationship), please avoid reporting these”

However, we agree with the reviewer that this is an absolutely critical question for future research. Future work may conduct a comprehensive investigation into how socio-demographic and environmental factors such as race and racism affect the development of personalized functional brain network topography, as we now suggest in the revised manuscript on lines 441-444:

“Future work may also expand upon our findings to explore other cognitive domains (e.g., social perception) that may be supported by distinct patterns of PFN topography as well as socio-demographic and environmental factors (e.g., socio-economic resources, race/ethnicity, and structural racism) that may shape the development of PFN topography.”

We also agree with the reviewer that analyses of biological sex are important. An independent study from our laboratory has explored this question in detail (Shanmugan et al., 2022). This work found that there are indeed sex differences in the functional topography of association networks (particularly the ventral attention, default mode, and fronto-parietal networks) in youth, and these sex differences were spatially correlated with the expression of genes on the X chromosome. In our revised manuscript on lines 288-292, we highlight this work as a key motivating factor for our inclusion of sex as a covariate in all of our models:

“We then applied linear mixed-effects models to probe the association between total cortical representation of each PFN and general cognition while accounting for age, sex (motivated by prior findings⁵⁶ that patterns of PFN total cortical representation differ by biological sex), family (to account for siblings in the ABCD dataset), and head motion (mean FD) as model covariates”

5. Why not test if functional topography predicts social perception? This would complement analyses to test for specificity of PFNs for cognition vs other psychological processes, as well as

inform future work regarding PFNs and social perception.

Although we agree that this would be an interesting avenue for future research, we are not able to assess predictions of social perception with the available behavioral data that was collected in the ABCD Study. We have made a note in the revised Discussion section that this would be an interesting idea for future studies on lines 441-442.

“Future work may also expand upon our findings to explore other cognitive domains (e.g., social perception) that may be supported by distinct patterns of PFN topography”

We also point out that the previous study (Cui et al., 2020, Neuron) whose main findings we have replicated and expanded upon here did investigate the relationship between PFN topography and social cognition, finding that prediction accuracy was relatively low (mean partial $r = 0.12$).

6. The effect sizes here are good but not great. Considering the need for translation of neuroscience and neuroimaging findings that can inform clinical decision making in child and adolescence mental health, I would like to see this issue explored in greater detail in the discussion, particularly in the context of predictive modeling, comparisons of effect sizes to other imaging studies using ABCD or comparable large-scale datasets and/or methodological approaches, and considerations as well as challenges for clinical translational potential of small to medium effect sizes in brain-wide association studies and fMRI.

We are happy to clarify. It should be emphasized that our main multivariate prediction results (shown in Figures 2, 3 and 4) have substantially larger effect sizes than have been found in prior work. Indeed, according to Marek, Tervo-Clemmens et al. (2022), effect sizes across three large datasets totaling over 35,000 scans show median correlation effect sizes between 0.02 to 0.03 with a range of less than -0.2 to 0.2. Our main findings (with effect sizes exceeding 0.4, see Figure 2) are greater than would be expected in a typical analysis with a dataset of this size. Additionally, it is worth pointing the reported effect size of the multivariate analyses is nearly identical to our prior work (Cui et al., 2020), despite the fact that due to the “winner’s curse” replication analyses typically show smaller effect sizes than original studies (Patil et al., 2016). In the revised manuscript, we address the reviewer’s comment about effect sizes in five ways:

First, we have restructured our Results section to first focus on the predictive modeling results followed by the univariate association results.

Second, we have reduced the text in the discussion section that focus on our univariate association analyses, instead focusing more on our predictive modeling results which have much larger effect sizes on lines 327-332:

“In the largest study to use precision functional brain mapping to investigate cognition in children to date, we found reproducible associations between individual differences in

functional brain network organization and individual differences in cognition. Replicating key findings from a prior study¹⁶ in samples that were an order of magnitude larger, we trained cross-validated models on the complex multivariate pattern of personalized functional network topography to predict individual differences in cognitive functioning in unseen participants' data."

Third, we contextualize the small effect sizes observed in the mass-univariate association analysis by pointing out that they align with expected effect sizes for brain-behavior associations in datasets of this size and remain highly reproducible across multiple large samples (lines 301-304):

"We also note that although the effect sizes for these univariate associations are small, they fall within or above the expected range for accurately-estimated brain-behavior effect sizes in studies of this size¹⁷ and these effects are highly reproducible across studies and samples."

Fourth, we further contextualize the effect sizes observed in the multivariate prediction analyses by pointing out that they align with expected effect sizes for predictive modeling studies using functional connectivity on lines 175-179:

"We found that individualized functional topography accurately predicted out-of-sample cognitive performance in both samples (**Figure 2a**, discovery: $r = 0.41$, $p < 0.001$, 95% CI: [0.39, 0.44]; replication: $r = 0.45$, $p < 0.001$, 95% CI: [0.43, 0.48]), with effect sizes at the higher end of the expected range from predictive modeling studies using functional connectivity in prior work.⁴¹⁻⁴³"

Fifth, as suggested by the reviewer, we have added the following discussion point to address the importance of moderate to large effect sizes and generalizability to new samples for clinical translation on lines 360-362:

"In order to move toward the goal of supporting child and adolescent mental health, such moderate to large effect sizes and generalizability to new samples are essential for predictive models to have clinical utility."

7. How was the tuning parameter for the ridge regression determined?

We have clarified this point in the revised Methods section on lines 628-642:

"Our primary ridge regression models were trained and tested on the ABCD reproducible matched samples^{40,41} using nested two-fold cross-validation (2F-CV), with outer 2F-CV estimating the generalizability of the model and the inner 2F-CV determining the optimal tuning parameter (λ) for the ridge regression model. For the inner 2F-CV, one subset was selected to train the model under a given λ value in the range $[2^{10}, 2^9, \dots, 2^4, 2^5]$ (i.e., 16 values in total)¹⁶, and the remaining subset was used to test the model. This procedure was

repeated 2 times such that each subset was used once as the testing dataset, resulting in two inner 2F-CV loops in total. For each λ value, the correlation r between the actual and predicted outcome as well as the mean absolute error (MAE) were calculated for each inner 2F-CV loop, and then averaged across the two inner loops. The sum of the mean correlation r and reciprocal of the mean MAE was defined as the inner prediction accuracy, and the λ with the highest inner prediction accuracy was chosen as the optimal λ .¹⁶ Of note, the mean correlation r and the reciprocal of the mean MAE cannot be summed directly, because the scales of the raw values of these two measures are quite different. Therefore, we normalized the mean correlation r and the reciprocal of the mean MAE across all values and then summed the resultant normalized values.”

8. *Why not test the generalization of predictive models in the current study to Cui et al. 2020 (and vice versa)?*

While we agree that it would be interesting to test the exact same models from Cui et al., 2020 in this new dataset, several challenges related to differences between the datasets prohibit us from performing this analysis. Most notably, the outcome measures that we are predicting in each study come from different sets of behavioral tasks. Additionally, the features used to train the predictive models come from fMRI scans with different sequences and used different registration templates (fslr vs. fsaverage5). Given that both the precise feature sets and outcome variables were different across these two studies, it would not be appropriate to apply the same predictive models across studies. We have therefore clarified in the main text on lines 425-430 that our results highlight the *generalizability* of the findings in Cui et al., 2020 using both different data and different methods:

“Fourth, differences between the ABCD dataset and the dataset used in the original study¹⁶ (e.g., differences in scanning sequences, registration templates, and cognitive measures) prohibited us from directly applying the same models from the original study to this dataset directly. Our results therefore constitute a conceptual replication of the prior findings that demonstrates the robust generalizability of the results with both new data and new methods.”

9. *The authors state that “nonsignificant differences between participants in the discovery and replication samples were present across any socio-demographic variables, nor were there any significant differences in scores across the three cognitive domains.” Were there associations between socio-demographic variables and connectivity?*

First, as a point of clarification, we did not perform any analyses of connectivity in this study. However, we appreciate the reviewer’s point about the importance of investigating the role of socio-economic status (SES) in studies of cognition and brain development.

Prior studies using the ABCD dataset have rigorously explored associations among SES, cognitive functioning, and brain network properties (e.g., Ellwood-Lowe et al., 2021) and it has been consistently demonstrated that higher SES is associated with better performance on a wide range of cognitive tasks. In line with Reviewer 1’s suggestion (see Reviewer 1 Comment 1), we have conducted an additional analysis using Area Deprivation Index (ADI) as a measure of SES in our analyses of cognition prediction accuracy. Specifically, we trained two new sets of ridge regression models using the same procedure as in our main results to predict three domains of cognitive functioning (General Cognition, Executive Function, and Learning/Memory): the first set of models used only ADI as a predictor of cognitive functioning to reveal the baseline prediction accuracy that could be achieved, while the second set of models used ADI in addition to the multivariate pattern of personalized functional brain network (PFN) topography (as in our previous results). As shown in the new **Supplementary Table 1** copied below, we now highlight the differences in prediction accuracies across ridge regression models including vs. not including SES.

Prediction Accuracy	Discovery		Replication	
	r	p	r	p
General Cognition				
SES	0.26	1.35 x 10 ⁻⁵¹	0.28	4.77 x 10 ⁻⁵⁸
PFN Topography	0.41	3.05 x 10 ⁻¹⁴⁶	0.45	3.85 x 10 ⁻¹⁷⁴
SES + PFN Topography	0.43	1.01 x 10 ⁻¹⁵¹	0.46	1.80 x 10 ⁻¹⁷¹
Executive Function				
SES	0.07	1.14 x 10 ⁻⁴	0.09	3.25 x 10 ⁻⁷
PFN Topography	0.17	1.37 x 10 ⁻²³	0.16	5.48 x 10 ⁻²²
SES + PFN Topography	0.17	7.18 x 10 ⁻²²	0.17	2.59 x 10 ⁻²³
Learning/Memory				
SES	0.13	2.96 x 10 ⁻¹³	0.16	4.46 x 10 ⁻¹⁹
PFN Topography	0.27	2.06 x 10 ⁻⁶¹	0.27	2.91 x 10 ⁻⁵⁷
SES + PFN Topography	0.27	3.53 x 10 ⁻⁵⁷	0.27	2.35 x 10 ⁻⁵⁶

Supplementary Table 1. Predictive models incorporating socio-economic status (SES). Prediction accuracy, measured as the correlation *r* between actual and predicted cognitive performance, is shown for ridge regression models trained to predict cognitive performance across three domains (General Cognition, Executive Function, and Learning/Memory) across

both discovery and replication samples. Results from three sets of predictive models are shown: “SES” refers to models trained only on socio-economic status as measured by the areal deprivation index; “PFN Topography” refers to models trained on the multivariate pattern of personalized functional brain network (PFN) topography for each individual (as presented in the main text and in Figures 2 and 3); and “SES + PFN Topography” refers to models trained on both socio-economic status and PFN topography. Although SES is a significant predictor of cognitive functioning, models trained on PFN topography yield much stronger predictions of cognitive performance than SES alone, and the addition of SES information to models trained on PFN topography does not substantially increase prediction accuracy. This observation suggests that the spatial topography of individually-defined functional brain networks accounts for additional inter-individual variance in cognitive performance beyond what is accounted for by SES alone.

The modest prediction accuracies for the SES-only models suggest that, in line with prior studies, SES is a significant predictor of cognitive functioning. Comparing prediction accuracies across models, we find that models trained on PFN topography yield much stronger predictions of cognitive performance than SES alone, and that the addition of SES information to models trained on PFN topography does not substantially increase prediction accuracy. This observation suggests that SES is associated with some of the inter-individual heterogeneity in PFN topography but does not fully account for the association between PFN topography and cognitive performance. Put another way, the spatial topography of individually-defined functional brain networks accounts for additional inter-individual variance in cognitive performance beyond what is accounted for by SES alone. This finding is now described in the revised manuscript on lines 185-190:

“Given that many prior studies in this dataset have demonstrated associations between socio-economic status and cognitive functioning,⁴⁴⁻⁵³ we also note that our predictive models trained on PFN topography outperformed models trained on socio-economic status as measured by areal deprivation index (ADI) alone, and we observed little to no improvement in prediction accuracy when models were trained on both ADI and PFN topography together (**Supplementary Table 1**).”

To further demonstrate that the inclusion of ADI as a covariate does not alter our univariate association results, we conducted an additional sensitivity analysis now included in the new **Supplementary Table 3** copied below. This analysis confirmed that all of our univariate association results remained significant with the inclusion of this covariate, with general cognition still significantly associated with the total cortical representation of all three fronto-parietal PFNs across both the discovery and replication samples.

Predictors	PFN 3				PFN 15				PFN 17			
	β	Std. Error	t	p _{bonf}	β	Std. Error	t	p _{bonf}	β	Std. Error	t	p _{bonf}
Discovery												
Intercept	0.01	0.02	0.42	0.672	0.08	0.02	3.24	1.21 x 10⁻³	-0.04	0.02	-1.80	7.26e-02
Age	-0.04	0.02	-2.23	0.026	-0.03	0.02	-1.43	0.152	-0.05	0.02	-2.59	9.65 x 10⁻³
Sex	-0.05	0.04	-1.51	0.131	-0.16	0.03	-4.53	6.23 x 10⁻⁶	0.07	0.03	1.91	0.0557
Mean FD	0.12	0.02	7.07	1.83 x 10⁻¹²	0.12	0.02	6.65	3.36 x 10⁻¹¹	0.04	0.02	2.08	0.038
SES	-0.00	0.02	-0.14	0.892	0.01	0.02	0.51	0.610	0.02	0.02	1.34	0.179
General Cognition	0.08	0.03	3.20	1.40 x 10⁻³	0.08	0.03	3.10	1.96 x 10⁻³	0.11	0.03	4.06	5.09 x 10⁻⁵
Replication												
Intercept	0.01	0.03	0.44	0.661	0.07	0.03	2.62	8.86 x 10⁻³	-0.02	0.03	-0.78	0.436
Age	-0.02	0.02	-0.86	0.392	-0.06	0.02	-3.13	1.77 x 10⁻³	-0.05	0.02	-2.68	7.35 x 10⁻³
Sex	-0.05	0.04	-1.43	0.154	-0.15	0.04	-4.10	4.22 x 10⁻⁵	0.03	0.04	0.86	0.388
Mean FD	0.16	0.02	9.32	2.18 x 10⁻²⁰	0.09	0.02	4.84	1.37 x 10⁻⁶	0.04	0.02	2.41	0.016
SES	0.03	0.02	1.50	0.133	0.05	0.02	2.71	6.71 x 10⁻³	0.04	0.02	2.36	0.018
General Cognition	0.07	0.03	2.53	0.011	0.07	0.03	2.57	0.010	0.10	0.03	3.93	8.51 x 10⁻⁵

Supplementary Table 3. Sensitivity analyses controlling for socio-economic status (SES).

Linear mixed effects models associating general cognition with the total cortical representation of fronto-parietal PFNs remain significant across both the discovery and replication samples when controlling for socio-economic status (SES) as measured by areal deprivation index.

Further, to encourage future research studies to explore this important line of investigation in more depth, we have also included the following text in our revised manuscript on lines 441-444:

“Future work may also expand upon our findings to explore other cognitive domains (e.g., social perception) that may be supported by distinct patterns of PFN topography as well as socio-demographic and environmental factors (e.g., socio-economic resources, race/ethnicity, and structural racism) that may shape the development of PFN topography.”

10. The authors state that the findings represent “...a critical step toward understanding healthy neurocognitive development.” However, this study leveraged a dataset with a heterogenous sample of youth with varying levels of internalizing/externalizing symptoms (with ~40% in the clinically significant range). These findings don’t necessarily represent an entirely homogenous sample of unaffected, healthy control youth or “healthy” neurocognitive development. The authors may want to consider rephrasing.

We appreciate this comment and have accordingly rephrased this sentence on lines 336-339. To emphasize that our study doesn’t represent an entirely homogenous sample of unaffected, healthy control youth, we have removed the word “healthy” as suggested:

“Together, these findings demonstrate that the link between functional network topography and cognition in children on the precipice of the transition to adolescence is highly reproducible, representing a critical step toward understanding heterogeneity in neurocognitive development.”

Minor comments:

11. In the introduction and sections of the discussion, the authors could consider expanding more on the importance of inter-individual heterogeneity in network organization and predictive models, particularly due to the role of the frontoparietal network across domains of psychopathology.

We agree. We have now included the following text in the revised Introduction (lines 83-86) and the Discussion (lines 447-450):

“These studies have revealed substantial inter-individual heterogeneity in functional topography,²⁰⁻²⁵ with especially notable heterogeneity in networks in association cortex that support higher-order cognition and are implicated in cognitive impairments in psychiatric illness in adults.^{21,64}”

“Given that the functional topography of networks implicated in cognitive impairments and psychiatric illness (e.g., the fronto-parietal network⁶⁴) tend to have the highest inter-individual heterogeneity¹⁶, studies of personalized networks may be essential in better understanding these symptoms.”

12. Please specify what the variable “family” refers to (e.g., income, siblings, environment). This seems overly broad and should be clarified.

A brief definition of this variable is provided in the Methods section on line 616:

“a random intercept for family (accounting for siblings) and site groupings”

To ensure that this point is clear throughout the manuscript, we have added a second description in the results section on lines 290-292:

“These models accounted for age, sex (motivated by prior findings⁵⁶ that patterns of PFN total cortical representation differ by biological sex), family (to account for siblings in the ABCD dataset), and head motion (mean FD) as model covariates”

13. Table 2: what was the proportion of girls vs boys in the total and discovery/replication samples? This is not specified.

We have specified in the revised Table 2 copied below that the percentages shown for the Discovery and Replication samples refer to the proportion of females.

	Total	Discovery	Replication	P-value
Age (Months)	119.5 (±7.5)	119.5 (±7.6)	119.5 (±7.5)	0.94
Sex (F)	3,494 (50.1%)	1,806 (51.2%)	1,688 (49.0%)	0.062
General Cognition	0.1 (±0.7)	0.1 (±0.7)	0.1 (±0.7)	0.18
Executive Function	0.0 (±0.8)	0.0 (±0.8)	0.1 (±0.7)	0.23
Learning/Memory	0.1 (±0.7)	0.1 (±0.7)	0.0 (±0.7)	0.22
ADHD Medications	536 (7.7%)	283 (8.0%)	253 (7.3%)	0.30
Antidepressant Medication	122 (1.8%)	69 (2.0%)	53 (1.5%)	0.20
Antipsychotic Medication	40 (0.6%)	23 (0.7%)	17 (0.5%)	0.43
CBCL Ext.				0.93
Mean (SD)	4.2 (±5.5)	4.2 (±5.6)	4.1 (±5.4)	
Missing	1 (0.0%)	0 (0%)	1 (0.0%)	
CBCL Int.				0.50
Mean (SD)	5.0 (±5.5)	5.0 (±5.6)	4.9 (±5.4)	
Missing	1 (0.0%)	0 (0%)	1 (0.0%)	
CBCL Prob.				0.99
Mean (SD)	17.3 (±17.2)	17.4 (±17.5)	17.1 (±16.9)	
Missing	1 (0.0%)	0 (0%)	1 (0.0%)	
Household Income				0.28
[<50K]	1,645 (23.6%)	806 (22.9%)	839 (24.4%)	
[≥50K & <100K]	1,902 (27.3%)	983 (27.9%)	919 (26.7%)	
[≥100K]	2,870 (41.2%)	1,451 (41.2%)	1,419 (41.2%)	
Missing	552 (7.9%)	284 (8.1%)	268 (7.8%)	
Race				0.46
White	4,723 (67.8%)	2,426 (68.8%)	2,297 (66.7%)	
Black	910 (13.1%)	442 (12.5%)	468 (13.6%)	
Asian	145 (2.1%)	71 (2.0%)	74 (2.1%)	
AIAN/NHPI	41 (0.6%)	23 (0.7%)	18 (0.5%)	
Other	242 (3.5%)	114 (3.2%)	128 (3.7%)	
Mixed	823 (11.8%)	410 (11.6%)	413 (12.0%)	
Missing	85 (1.2%)	38 (1.1%)	47 (1.4%)	
Parent Education				0.31
< HS Diploma	236 (3.4%)	117 (3.3%)	119 (3.5%)	
HS Diploma/GED	542 (7.8%)	254 (7.2%)	288 (8.4%)	
Some College	1,781 (25.6%)	905 (25.7%)	876 (25.4%)	
Bachelor	1,885 (27.0%)	980 (27.8%)	905 (26.3%)	
Post Graduate Degree	2,519 (36.1%)	1,264 (35.9%)	1,255 (36.4%)	
Missing	6 (0.1%)	4 (0.1%)	2 (0.1%)	

Table 2. Demographic characteristics and variables of interest in the matched discovery ($n=3,525$) and replication ($n=3,447$) samples. Acronyms: AIAN = American Indian/Alaska Native; NHPI = Native Hawaiian and other Pacific Islander; HS = High School; GED = General Educational Development; CBCL = Child Behavior Checklist.

References

- Bijsterbosch, J. D., Valk, S. L., Wang, D., & Glasser, M. F. (2021). Recent developments in representations of the connectome. *NeuroImage*, *243*.
<https://doi.org/10.1016/j.neuroimage.2021.118533>
- Ciric, R., Rosen, A. F. G., Erus, G., Cieslak, M., Adebimpe, A., Cook, P. A., Bassett, D. S., Davatzikos, C., Wolf, D. H., & Satterthwaite, T. D. (2018). Mitigating head motion artifact in functional connectivity MRI. *Nature Protocols*, *13*(12), 2801–2826.
<https://doi.org/10.1038/s41596-018-0065-y>
- Ciric, R., Wolf, D. H., Power, J. D., Roalf, D. R., Baum, G. L., Ruparel, K., Shinohara, R. T., Elliott, M. A., Eickhoff, S. B., Davatzikos, C., Gur, R. C., Gur, R. E., Bassett, D. S., & Satterthwaite, T. D. (2017). Benchmarking of participant-level confound regression strategies for the control of motion artifact in studies of functional connectivity. *NeuroImage*, *154*, 174–187. <https://doi.org/10.1016/j.neuroimage.2017.03.020>
- Cui, Z., Li, H., Xia, C. H., Larsen, B., Adebimpe, A., Baum, G. L., Cieslak, M., Gur, R. E., Gur, R. C., Moore, T. M., Oathes, D. J., Alexander-Bloch, A. F., Raznahan, A., Roalf, D. R., Shinohara, R. T., Wolf, D. H., Davatzikos, C., Bassett, D. S., Fair, D. A., ... Satterthwaite, T. D. (2020). Individual Variation in Functional Topography of Association Networks in Youth. *Neuron*, *106*(2), 340-353.e8.
<https://doi.org/10.1016/j.neuron.2020.01.029>

Dosenbach, N. U. F., Koller, J. M., Earl, E. A., Miranda-Dominguez, O., Klein, R. L., Van, A. N., Snyder, A. Z., Nagel, B. J., Nigg, J. T., Nguyen, A. L., Wesevich, V., Greene, D. J., & Fair, D. A. (2017). Real-time motion analytics during brain MRI improve data quality and reduce costs. *NeuroImage*, *161*, 80–93.

<https://doi.org/10.1016/j.neuroimage.2017.08.025>

Ellwood-Lowe, M. E., Whitfield-Gabrieli, S., & Bunge, S. A. (2021). Brain network coupling associated with cognitive performance varies as a function of a child's environment in the ABCD study. *Nature Communications*, *12*(1). <https://doi.org/10.1038/s41467-021-27336-y>

Fortin, J.-P., Cullen, N., Sheline, Y. I., Taylor, W. D., Aselcioglu, I., Cook, P. A., Adams, P., Cooper, C., Fava, M., McGrath, P. J., McInnis, M., Phillips, M. L., Trivedi, M. H., Weissman, M. M., & Shinohara, R. T. (2018). Harmonization of cortical thickness measurements across scanners and sites. *NeuroImage*, *167*, 104–120.

<https://doi.org/10.1016/j.neuroimage.2017.11.024>

Fortin, J.-P., Parker, D., Tunç, B., Watanabe, T., Elliott, M. A., Ruparel, K., Roalf, D. R., Satterthwaite, T. D., Gur, R. C., Gur, R. E., Schultz, R. T., Verma, R., & Shinohara, R. T. (2017). Harmonization of multi-site diffusion tensor imaging data. *NeuroImage*, *161*, 149–170. <https://doi.org/10.1016/j.neuroimage.2017.08.047>

Garthwaite, P. H. (1994). An Interpretation of Partial Least Squares. *Journal of the American Statistical Association*, *89*(425), 122–127.

<https://doi.org/10.1080/01621459.1994.10476452>

Gonzalez, M. R., Palmer, C. E., Uban, K. A., Jernigan, T. L., Thompson, W. K., & Sowell, E. R. (2020). Positive Economic, Psychosocial, and Physiological Ecologies Predict Brain

Structure and Cognitive Performance in 9–10-Year-Old Children. *Frontiers in Human Neuroscience*, 14, 578822. <https://doi.org/10.3389/fnhum.2020.578822>

Gratton, C., Laumann, T. O., Nielsen, A. N., Greene, D. J., Gordon, E. M., Gilmore, A. W., Nelson, S. M., Coalson, R. S., Snyder, A. Z., Schlaggar, B. L., Dosenbach, N. U. F., & Petersen, S. E. (2018). Functional Brain Networks Are Dominated by Stable Group and Individual Factors, Not Cognitive or Daily Variation. *Neuron*, 98(2), 439-452.e5. <https://doi.org/10.1016/j.neuron.2018.03.035>

Hermosillo, R. J. M., Moore, L. A., Fezcko, E., Dworetzky, A., Pines, A., Conan, G., Mooney, M. A., Randolph, A., Adeyemo, B., Earl, E., Perrone, A., Carrasco, C. M., Uriarte-Lopez, J., Snider, K., Doyle, O., Cordova, M., Nagel, B. J., Ewing, S. W. F., Satterthwaite, T., ... Fair, D. A. (2022). *A Precision Functional Atlas of Network Probabilities and Individual-Specific Network Topography* (p. 2022.01.12.475422). bioRxiv. <https://doi.org/10.1101/2022.01.12.475422>

Kirlic, N., Colaizzi, J. M., Cosgrove, K. T., Cohen, Z. P., Yeh, H.-W., Breslin, F., Morris, A. S., Aupperle, R. L., Singh, M. K., & Paulus, M. P. (2021). Extracurricular Activities, Screen Media Activity, and Sleep May Be Modifiable Factors Related to Children's Cognitive Functioning: Evidence From the ABCD Study®. *Child Development*, 92(5), 2035–2052. <https://doi.org/10.1111/cdev.13578>

Patil, P., Peng, R. D., & Leek, J. T. (2016). What Should Researchers Expect When They Replicate Studies? A Statistical View of Replicability in Psychological Science. *Perspectives on Psychological Science: A Journal of the Association for Psychological Science*, 11(4), 539–544. <https://doi.org/10.1177/1745691616646366>

Power, J. D., Schlaggar, B. L., Lessov-Schlaggar, C. N., & Petersen, S. E. (2013). Evidence for Hubs in Human Functional Brain Networks. *Neuron*, *79*(4), 798–813.

<https://doi.org/10.1016/j.neuron.2013.07.035>

Rosenberg, M. D., & Finn, E. S. (2022). How to establish robust brain-behavior relationships without thousands of individuals. *Nature Neuroscience*, *25*(7), 835–837.

<https://doi.org/10.1038/s41593-022-01110-9>

Satterthwaite, T. D., Ciric, R., Roalf, D. R., Davatzikos, C., Bassett, D. S., & Wolf, D. H. (2019). Motion artifact in studies of functional connectivity: Characteristics and mitigation strategies. *Human Brain Mapping*, *40*(7), 2033–2051. <https://doi.org/10.1002/hbm.23665>

Satterthwaite, T. D., Elliott, M. A., Gerraty, R. T., Ruparel, K., Loughead, J., Calkins, M. E., Eickhoff, S. B., Hakonarson, H., Gur, R. C., Gur, R. E., & Wolf, D. H. (2013). An improved framework for confound regression and filtering for control of motion artifact in the preprocessing of resting-state functional connectivity data. *NeuroImage*, *64*, 240–256. <https://doi.org/10.1016/j.neuroimage.2012.08.052>

Satterthwaite, T. D., Wolf, D. H., Loughead, J., Ruparel, K., Elliott, M. A., Hakonarson, H., Gur, R. C., & Gur, R. E. (2012). Impact of in-scanner head motion on multiple measures of functional connectivity: Relevance for studies of neurodevelopment in youth.

NeuroImage, *60*(1), 623–632. <https://doi.org/10.1016/j.neuroimage.2011.12.063>

Satterthwaite, T. D., Wolf, D. H., Ruparel, K., Erus, G., Elliott, M. A., Eickhoff, S. B., Gennatas, E. D., Jackson, C., Prabhakaran, K., Smith, A., Hakonarson, H., Verma, R., Davatzikos, C., Gur, R. E., & Gur, R. C. (2013). Heterogeneous impact of motion on fundamental patterns of developmental changes in functional connectivity during youth. In

NeuroImage (Vol. 83). Elsevier Science.

<https://doi.org/10.1016/j.neuroimage.2013.06.045>

Shanmugan, S., Seidlitz, J., Cui, Z., Adebimpe, A., Bassett, D. S., Bertolero, M. A., Davatzikos, C., Fair, D. A., Gur, R. E., Gur, R. C., Larsen, B., Li, H., Pines, A., Raznahan, A., Roalf, D. R., Shinohara, R. T., Vogel, J., Wolf, D. H., Fan, Y., ... Satterthwaite, T. D. (2022).

Sex differences in the functional topography of association networks in youth.

Proceedings of the National Academy of Sciences of the United States of America,

119(33), e2110416119. <https://doi.org/10.1073/pnas.2110416119>

Shoval, G., Visoki, E., Moore, T. M., DiDomenico, G. E., Argabright, S. T., Huffnagle, N. J., Alexander-Bloch, A. F., Waller, R., Keele, L., Benton, T. D., Gur, R. E., & Barzilay, R.

(2021). Evaluation of Attention-Deficit/Hyperactivity Disorder Medications,

Externalizing Symptoms, and Suicidality in Children. *JAMA Network Open*, 4(6),

e2111342. <https://doi.org/10.1001/jamanetworkopen.2021.11342>

Sydnor, V. J., Larsen, B., Bassett, D. S., Alexander-Bloch, A., Fair, D. A., Liston, C., Mackey,

A. P., Milham, M. P., Pines, A., Roalf, D. R., Seidlitz, J., Xu, T., Raznahan, A., &

Satterthwaite, T. D. (2021). Neurodevelopment of the association cortices: Patterns, mechanisms, and implications for psychopathology. *Neuron*, 109(18), 2820–2846.

<https://doi.org/10.1016/j.neuron.2021.06.016>

Thompson, R. C., Montena, A. L., Liu, K., Watson, J., & Warren, S. L. (2022). Associations of

Family Distress, Family Income, and Acculturation on Pediatric Cognitive Performance

Using the NIH Toolbox: Implications for Clinical and Research Settings. *Archives of*

Clinical Neuropsychology: The Official Journal of the National Academy of

Neuropsychologists, 37(4), 798–813. <https://doi.org/10.1093/arclin/acab082>

Thompson, W. K., Barch, D. M., Bjork, J. M., Gonzalez, R., Nagel, B. J., Nixon, S. J., & Luciana, M. (2019). The structure of cognition in 9 and 10 year-old children and associations with problem behaviors: Findings from the ABCD study's baseline neurocognitive battery. *Developmental Cognitive Neuroscience, 36*, 100606.
<https://doi.org/10.1016/j.dcn.2018.12.004>

Tomasi, D., & Volkow, N. D. (2021). Associations of family income with cognition and brain structure in USA children: Prevention implications. *Molecular Psychiatry, 26*(11), Article 11. <https://doi.org/10.1038/s41380-021-01130-0>

REVIEWER COMMENTS

Reviewer #3 (Remarks to the Author):

I find the authors response to my critiques satisfactory.

Reviewer #4 (Remarks to the Author):

The authors have addressed this reviewer's critiques. Additional supplemental analyses would be helpful for addressing the discrepancies between performances of predictive models and weak associations raised by Reviewer 2. Overall, the manuscript is greatly improved. The authors were also thorough in addressing reviewer concerns.

REVIEWER COMMENTS

Reviewer #3 (Remarks to the Author):

I find the authors response to my critiques satisfactory.

We thank the reviewer for their approval of the manuscript.

Reviewer #4 (Remarks to the Author):

The authors have addressed this reviewer's critiques. Additional supplemental analyses would be helpful for addressing the discrepancies between performances of predictive models and weak associations raised by Reviewer 2. Overall, the manuscript is greatly improved. The authors were also thorough in addressing reviewer concerns.

We appreciate this positive appraisal and are happy to clarify regarding the differential effect sizes between our mass-univariate association analyses and multivariate prediction analyses. These two analyses make use of both different input features and analysis methodology that yields a larger effect size for the multivariate analysis. While the univariate association analyses make use of a simple summary metric (total cortical representation; one value per network), the multivariate predictive modeling analyses make use of the full richness and complexity of vertex-wise functional topography. Accordingly, the univariate analysis considers only one feature per network, whereas the multivariate analyses can draw nearly 60,000 features per network as part of a machine-learning approach to this high-dimensional data (see schematic in Figure 1).

It is therefore expected that our multivariate analyses have larger effect sizes than the simple univariate association analyses, given the use of the far richer feature set. To clarify this point in the revised manuscript, we have added the following sentences to lines 666-670:

“We further note that our multivariate ridge-regression analyses leverage the full richness and complexity of PFN topography across vertices, while our univariate linear mixed effects analyses of total cortical representation use a single scalar-value summary statistic (total cortical representation) per PFN. Thus, effect sizes for these multivariate prediction analyses are expected to be larger than effect sizes for our univariate association analyses.”

Regarding the split-half reliability analyses, we would like to underscore that the reliability of person-specific networks has been extensively validated in prior work (Gordon et al. 2017 *Neuron*; Glasser et al., 2016 *Nature*; Laumann et al., 2015 *Neuron*; Wang et al., 2015, *Nature Neuroscience*; Hacker et al., 2013, *NeuroImage*; Birn et al., 2015, *NeuroImage*). This approach has been broadly applied to studies of brain development (e.g. Pines et al., 2022 *Nature Communications*; Cui et al., 2022 *Biological Psychiatry*; Shanmugan et al., 2022 *PNAS*; Cui et al., 2020 *Neuron*).

Moreover, in response to prior reviewer feedback, we conducted reliability analyses in a subset of ten participants who had a large quantity low-motion quality data for this analysis. It has been shown in prior work (Gordon et al. 2017 *Neuron*; Laumann et al., 2015 *Neuron*) that the derivation of reliable precision functional brain maps requires substantial fMRI data quantity, with ten minutes considered a minimum. In our split-half reliability analysis, we were able to take advantage of having ten participants with at least twenty minutes of low-motion resting-state data which we could then split into two ten-minute halves to derive highly reliable personalized functional brain maps. We explain this point in the revised manuscript on lines 594-597:

“Split-half reliability of the PFN loadings were assessed in ten participants who had the longest duration of low-motion quality resting-state data exceeding 20 minutes allowing us to derive PFNs in two 10-minute segments, as previously described in prior work,⁹⁶ given the necessity of sufficient scan duration for the derivation of precision functional networks.^{20,21}”

To further address this point, we also added a note describing how our split-half reliability results align with what has been shown in prior studies. These studies provide strong evidence of reliability with convergent results to complement the excellent split-half reliability we report in our sample. This point may be found in the revised manuscript on lines 597-600:

“This analysis revealed high intraclass correlation coefficients for PFN loadings across all 17 networks (ICCs: 0.84–0.99) indicating excellent reliability of this measure (**Supplementary Figure 4**) that aligns with what has been found in prior work.^{20,21}”

Finally, we note that computing personalized functional networks twice per participant (using a split-half approach) in thousands of participants would take many months of compute time even using one of the largest imaging-dedicated high-performance compute clusters in the world (CUBIC, at Penn). It is beyond the scope of the current work.

REVIEWERS' COMMENTS

Reviewer #4 (Remarks to the Author):

The authors have addressed all reviewer critiques.

REVIEWERS' COMMENTS

Reviewer #4 (Remarks to the Author):

The authors have addressed all reviewer critiques.

We thank the reviewers for their thoughtful comments which have greatly strengthened the manuscript.